# REAL-TIME REASONING AGENTS IN EVOLVING ENVIRONMENTS

**Yule Wen**[1*]    **Yixin Ye**[2*]   **Yanzhe Zhang**[3]    **Diyi Yang**[4]    **Hao Zhu**[4]
[1]Tsinghua University    [2]Shanghai Jiao Tong University
[3]Georgia Institute of Technology    [4]Stanford University

## ABSTRACT

Agents in the real world must make not only logical but also *timely* judgments. This requires continuous awareness of the dynamic environment: hazards emerge, opportunities arise, and other agents act, while the agent's reasoning is still unfolding. Despite advances in language model reasoning, existing approaches fail to account for this dynamic nature. We introduce *real-time reasoning* as a new problem formulation for agents in evolving environments and build **Real-Time Reasoning Gym** to demonstrate it. We study two paradigms for deploying language models in agents: (1) reactive agents, which employ language models with *bounded reasoning computation for rapid responses*, and (2) planning agents, which allow *extended reasoning computation for complex problems*. Our experiments show that even state-of-the-art models struggle with making logical and timely judgments in either paradigm. To address this limitation, we propose **AgileThinker**, which simultaneously engages *both reasoning paradigms*. AgileThinker consistently outperforms agents engaging only one reasoning paradigm as the task difficulty and time pressure rise, effectively balancing reasoning depth and response latency. Our work establishes real-time reasoning as a critical testbed for developing practical agents and provides a foundation for research in temporally constrained AI systems, highlighting a path toward real-time capable agents.

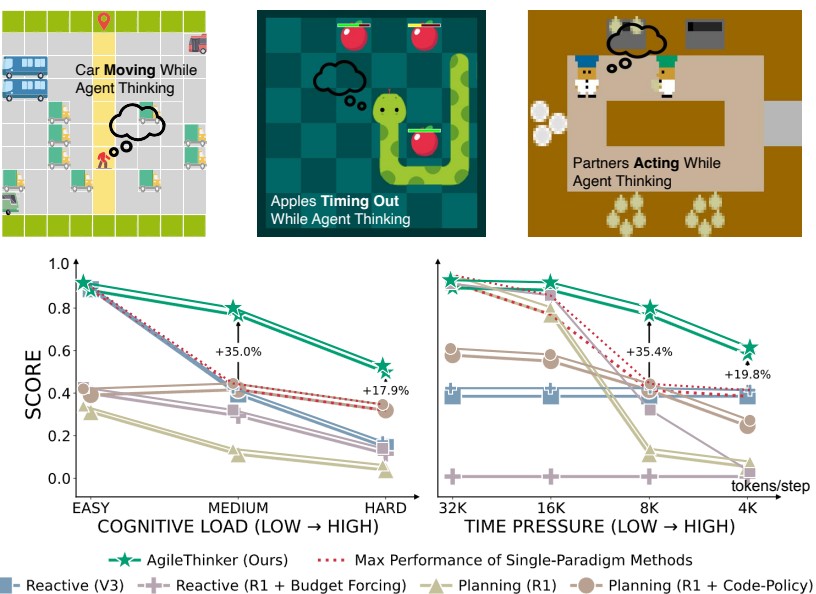

Figure 1: Upper: Three real-time games, *Freeway*, *Snake*, and *Overcooked*. Lower: Under *cognitive load* and *time pressure*, AgileThinker, engaging both *reactive* and *planning* reasoning, consistently outperforms agents that engage either of them. Scores are averaged across different games.

---

*Co-leading authors.

# 1 INTRODUCTION

Remember your first highway drive? White-knuckling the wheel, fixated on the car ahead. Then suddenly your exit appears and you're three lanes over. This captures how our brains work under pressure: reacting intuitively keeps us safe moment-to-moment, but deliberate planning gets us where we need to go. This example highlights a fundamental challenge: real-time reasoning (Stanovich & West, 2000; Kahneman, 2011; Evans & Stanovich, 2013). Under time pressures, agents must simultaneously support timely reactions and cognitively demanding long-term planning. Remarkably, humans excel at this balancing act: shifting seamlessly between instinct and analysis as situations demand. Expert drivers eventually handle both tasks effortlessly.

However, current Large Language Model (LLM)-based agents fail to live up to this challenge. Most existing work assume that the environments only change when the agents issue an action, ignoring the *dynamic* nature of the world, which evolves in parallel to the agent's computation. As a result, despite great effort in improving agent planning with LLM reasoning, including Yao et al. (2022); Gou et al. (2023); Putta et al. (2024); Ferrag et al. (2025), under this assumption, how to evaluate and improve the capability to make timely decisions is still an open question.

To bridge this gap, we introduce **Real-Time Reasoning Gym**, the first environment for language agents to reason in dynamic environments (§2). Our gym consists of three real-time games: *Freeway*, inspired by the Atari game under the same name, *Snake*, an adaptation of a popular game, and *Overcooked*, a two-player version of the collaborative video game. In each game, the state updates at a fixed rate regardless of whether the agent finishes its reasoning, and if no action is produced by the agent, a default action is used, simulating reasoning and acting in a real-time world. They challenge agents with different aspects of a dynamic environment: Freeway features dynamic hazards with moving cars, Snake involves dynamic opportunities as food appears and disappears, and Overcooked requires coordination with dynamic partners who act on their own. Real-Time Reasoning Gym is useful for studying different agent designs for real-time tasks.

To compare different design choices of real-time reasoning agents, we study two paradigms: reactive agents and planning agents (§3). Reactive agents ensure responsiveness by limiting computation, while planning agents are allowed to perform more extensive thinking. However, neither of them is perfect: planning agents cannot easily react to changes in the environment, and reactive agents fail to make strategic decisions. We propose **AgileThinker** (§3), a simple yet effective method that combines the strengths of *both paradigms*. Unlike agents with one paradigm that must choose between speed and accuracy, AgileThinker runs two LLMs in two parallel threads: a *planning thread* performs extended reasoning over frozen game states, and a *reactive thread* outputs timely decisions within environmental update time. Specifically, the reactive thread can reference partial reasoning traces from the ongoing planning process, enabling informed real-time decisions without waiting for complete analysis. This also differs from prior dual-system methods (Zhang et al., 2025; Liu et al., 2024; Christakopoulou et al., 2024), where either two systems operate independently, or one must wait for another to complete before accessing its outputs.

In this paper, we study the following research questions:

**RQ1** How do environment factors affect performance of agents in Real-Time Reasoning Gym?

**RQ2** How to balance reaction and planning resources in AgileThinker?

**RQ3** How well do the results we get with simulation in Real-Time Reasoning Gym match real-world walltime experiments?

To study these questions, we manipulate the cognitive load and time pressure of the games, facilitating systematic evaluation across both dimensions. We evaluate different design choices for reactive and planning agents, including budget forcing (Muennighoff et al., 2025) for reactive agents, code-as-a-policy (Liang et al., 2022) for planning agents. For fair comparison across agent designs, we use one model family, focusing on DeepSeek V3 and R1 because they are open-source and provide transparent reasoning trajectories required by AgileThinker. To assess generality, we also experiment with proprietary models (App. § C.3) and observe similar performance trend. Our results demonstrate that AgileThinker consistently outperforms single-paradigm methods by effectively balancing reactive and planning processes. This advantage is confirmed to translate to real-world scenarios through wall-clock time experiments. Ultimately, our work establishes a foundation for

developing language agents capable of complex reasoning under time constraints, bringing AI systems closer to human-level performance in dynamic environments.

## 2    REAL-TIME REASONING GYM

To evaluate the real-time reasoning capabilities of agents, we need an environment that is (1) dynamic: the environment state continuously updates without waiting for the agent's decision; (2) cognitively challenging: the tasks should be challenging enough so that logical reasoning is needed; (3) reproducible: simulated environments to avoid non-negligible noise from hardware factors.

To achieve these three desiderata, we consider a new formulation of decision-making problem. Unlike conventional turn-based environments, where the environment steps only after the agent finishes thinking and produces an action (Figure 2 left), in Real-Time Reasoning Gym, the environment steps forward at a fixed rate, even when the agent has not finished thinking. If no action is produced in time, a default action is applied (Figure 2 right). This simulates the real-world situation where the environment does not delay or accelerate according to the agent's computation time.

**Games**  In order to control the dynamics of the environment for evaluating real-time reasoning, we use real-time games in our gym. We created three games to capture different challenges that a dynamic environment brings: maintaining safety when hazards happen, seizing transient opportunities, and coordinating with partners (Tab. 1).

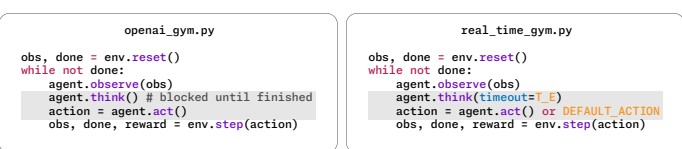

Figure 2:  Agent loops in OpenAI Gym (Brockman et al., 2016) and Real-Time Reasoning Gym.  Constants T_E and DEFAULT_ACTION will be explained in the following 'time pressure' paragraph.

In Freeway, the agent traverses multiple lanes of bidirectional traffic, which requires constant monitoring for oncoming cars while planning future trajectories to avoid becoming trapped mid-road. In Snake, the agent eats apples which are only available for a short period of time, creating opportunities that must be seized quickly, while greedy food collection might lead to positions where the growing snake traps itself. In Overcooked, the agent collaborates with a scripted partner following a non-stationary policy (App. § A). Efficient dish preparation requires not only planning a sequence of actions, but also coordinating effectively with the partner.

**Cognitive Load**  To systematically control how challenging the games are, we make the difficulty of each game tunable through a cognitive load factor (Tab. 1). In Freeway, difficulty is determined by the minimum number of steps required to traverse the road, since longer paths typically introduce more detours and require deeper planning horizon.  In Snake, we vary the density of obstacles, increasing route complexity and the need for look-ahead. In Overcooked, complexity is controlled by the length of an internal kitchen counter, as a longer counter expands navigation complexity and stretches temporal windows for high-level goals, creating larger discrepancies in long-term planning and immediate execution. For each game, we design 3 levels of difficulties, easy, medium, and hard; the corresponding ranges of each level can be found in App. Tab. 5.

**Time Pressure**   To enable reproducible and hardware-agnostic evaluation, we use token count as a time proxy to simulate the games, leveraging the fact that LLM decoding time scales almost linearly with output length through time-per-output-token (TPOT), while prefilling time becomes negligible for long sequences. This yields decoding time $T = N_T \times \text{TPOT}$, where $N_T$ is the generated token

Table 1: Different Games in Real-Time Reasoning Gym.

| Game | Dynamic Aspect | Cognitive Load Factor | Evaluation Metrics |
| --- | --- | --- | --- |
| Freeway | Hazards | Min steps to finish: $S$ | #Steps the agent takes to get to the other side |
| Snake | Opportunities | #Obstacles: $N$ | #Apples the agent eats before collision |
| Overcooked | Partners | Kitchen Counter Len.: $L$ | #Orders completed cooperatively |

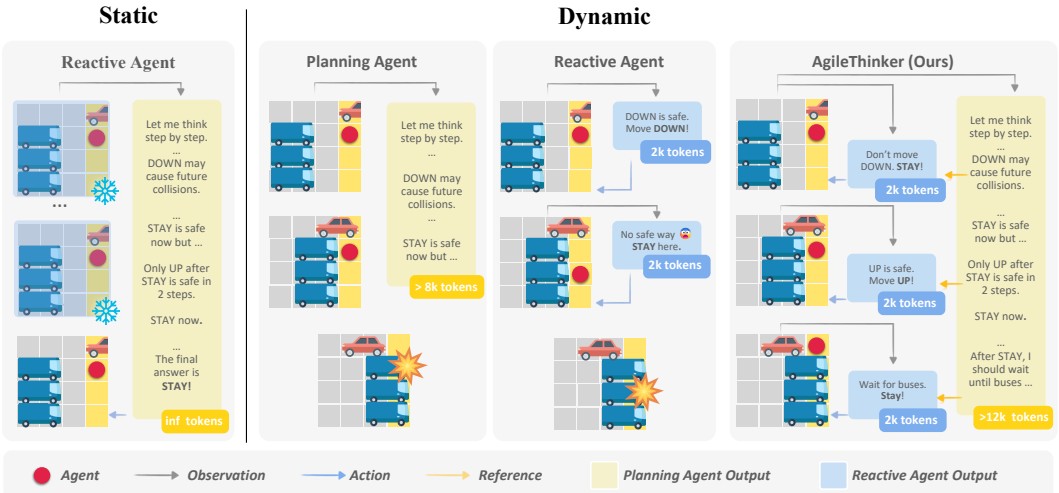

Figure 3: Existing evaluation setups for LLM Agents often assume a *static* setting, where the environment halts while the agent completes reasoning with unlimited computation. In Real-Time Reasoning Gym, environments are *dynamic*, evolving regardless of agents' computation state. As illustrated in the *Freeway* setting, **Planning Agent**, which performs extended reasoning without interruption, fails to act timely while **Reactive Agent**, which performs reasoning strictly within environment update period, lacks foresight and collides. **AgileThinker** combines both timely reaction and long-term planning to navigate such environments effectively.

count, allowing fair comparison across deployment scenarios. We impose time pressure by letting the environment step every $N_{T_\mathcal{E}}$, or T_E in Fig. 2, tokens generated by agents. When the agent cannot produce a valid action, we let the environment step with a DEFAULT_ACTION (Fig. 2). In Freeway and Snake, the default action is moving in the same direction as before, and in Overcooked, the default action is to stay idle. We consider four different time pressure levels, 32k, 16k, 8k, and 4k tokens per step, from low to high pressure. As shown in Figure 3, unlike existing static evaluations, the introduction of time pressure simulates the real world dynamic environments where the world does not freeze during agent reasoning.

**Evaluation**   Evaluation metric differs for each game. In Freeway, we evaluate the number of steps the agent takes to get to the other side, while the agent is reset to the origin every time it gets hit by a car; in Snake, we count the number of apples that the agent eats before a collision; and in Overcooked, we use the number of orders that the agent and the partner completed in total. These evaluation metrics represent the capability of the agents to solve the tasks not only logically but also timely. For each game, we normalize the scores by the highest score the agent could get in that game, so we always have a score between 0 and 1. As cognitive load and time pressure increase, we expect the scores decrease. However, the scores of an agent with strong real-time reasoning capabilities should decrease slower. It is worth noting that our gym is used to evaluate *design choices of agent systems* when the model or model family (e.g. DeepSeek-V3 and R1) is fixed. Therefore, cross-model comparisons may be unfair due to their different tokenizers and underlying architectures.

## 3   REAL-TIME REASONING AGENTS

To address the real-time reasoning problem, we consider two solution paradigms: (1) reactive, where the agent produces a new action at every environment step, and (2) planning, where the agent reasons across multiple steps to generate an action plan, which is then executed until the agent resumes reasoning. In the following, we discuss how to create agents following each of the two paradigms, and how these two paradigms are engaged in AgileThinker.

**Reactive agents**   We constrain reactive agents by a token budget $N_i$, ensuring they can respond within each environment update when $N_i \leq N_{T_\mathcal{E}}$. We consider two kinds of language models

for reactive agents: (1) non-thinking models[1] that produce limited tokens for each response; and (2) thinking models that produce extended reasoning which is cut off at the token budget $N_i$ with budget forcing (Muennighoff et al., 2025). In both cases, the agent produces one action per environment step, enabling immediate reaction to any change. This reactive approach is commonly used in agent systems these days.

**Planning agents**  While reactive agents ensure timeliness, their token budget restricts the complexity of reasoning they can perform within a single step. In contrast, planning agents can consider their plans through multiple environment steps. We consider two formats of plans: (1) multi-step actions where a thinking model is used to generate a sequence of multiple actions to be executed; and (2) code plans where a thinking model is used to generate a code snippet that automatically produces actions based on observation input (Liang et al., 2022; Zhang et al., 2025). Although (1) is often easier to generate, (2) is more adaptive to potential changes. Both formats allow for more deliberate, long-horizon decision-making by leveraging extended reasoning.

**AgileThinker**  All agents introduced above must complete their reasoning process before taking any action. To overcome this limitation, we propose AgileThinker, which employs two parallel threads to achieve both timely action generation and uninterrupted deep planning. The planning thread $\mathcal{P}$ runs an LLM that streams the thinking process for a multi-step action plan. Upon initialization, a reasoning process begins that continues until the execution of a plan. $\mathcal{P}$ cannot keep up with environmental changes (e.g. Steps 1-2 in Fig. 4). However, since its planning focuses on long-term

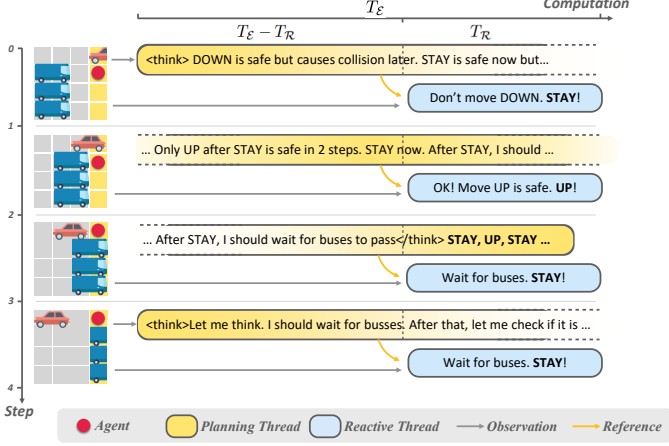

Figure 4: Two parallel threads in AgileThinker

objectives, many generated insights remain useful over extended horizons. In contrast, the reactive thread $\mathcal{R}$ runs a separate LLM under strict time constraints $T_{\mathcal{R}} \leq T_{\mathcal{E}}$, making decisions based on the latest observation and the (partial) output of $\mathcal{P}$ (see gray and yellow arrows in Fig. 4).

The coordination between the two threads $\mathcal{R}$ and $\mathcal{P}$ follows a time-sharing protocol: during each environment step, $\mathcal{P}$ operates continuously while $\mathcal{R}$ activates only in the final $T_{\mathcal{R}}$ time units. The hyperparameter $T_{\mathcal{R}}$ controls the resource trade-off between the two thread. With a larger $T_{\mathcal{R}}$, the reactive thread can be more adaptive, but there will be less reasoning from the planning thread to refer to. Effectively balancing planning and reaction resources is the key to success in AgileThinker, which will be discussed in §5.

## 4    IS SINGLE PARADIGM ENOUGH FOR REAL-TIME REASONING?

**Evaluation Setup:** To investigate how **cognitive load** and **time pressure** affect the performance respectively, we conduct two series of experiments. (1) Cognitive load varies (Easy, Medium, Hard) while time pressure is fixed at 8k tokens/step—lenient enough for non-thinking models to complete their responses, yet restrictive for thinking models. The intrinsic bound $N_i$ (see Section 3) for reactive agent is set to 8k. (2) Time pressure varies ($N_{T_{\mathcal{E}}} \in \{4k, 8k, 16k, 32k\}$) with medium cognitive load. Here, $N_i$ is set to 4k to ensure it remains lower than time pressure budget. We evaluate each agent 32 times (8 game seeds × 4 LLM sampling seeds) under each setting and report

---

[1]Thinking models are the LLMs trained with reinforcement learning to incentivize reasoning before generating answers (DeepSeek-AI et al., 2025), while non-thinking models are the LLMs that have not been specifically trained to generate long reasoning.

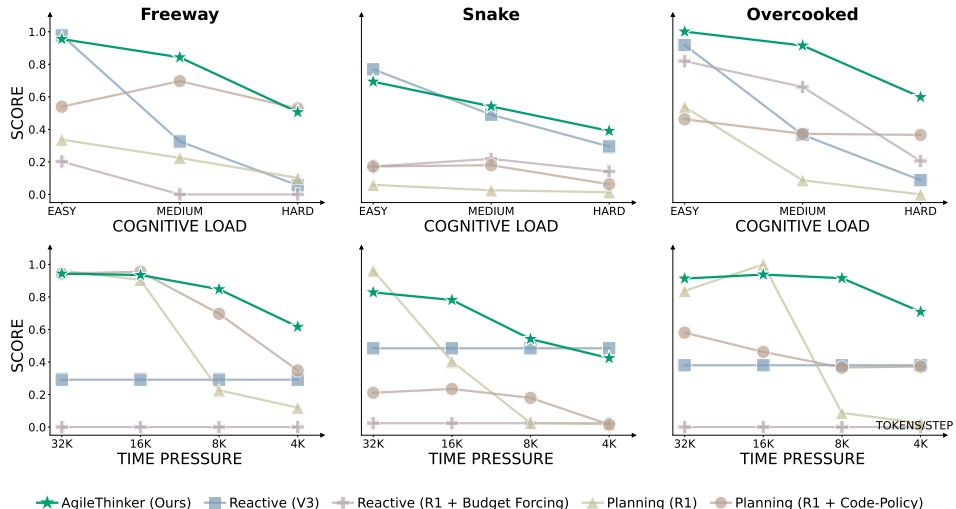

Figure 5: Performance of reasoning agents in Real-Time Reasoning Gym under varying cognitive loads and time pressures. Upper: we fix time pressure at 8k tokens per step and vary cognitive load. Lower: we fix cognitive load at medium level and vary time pressure. Full data at App. § C.

the average score of these samples. Details of the environments and score calculation can be found in App. § A, and prompts are provided in App. § B.

Figure 1 reports the average scores over three games, while Figure 5 provides a breakdown per game. The results show that reactive and planning agents fail to balance decision quality and efficiency, whereas our AgileThinker achieves robust performance under varying conditions.

**Reactive agent sacrifices decision quality for efficiency.** By design, the reactive agent restricts computation time less than $T_{\mathcal{E}}$ and maintains consistent performance across all time pressures. However, the limit on test-time scaling also causes a dramatic performance drop as cognitive load increases (scores falling from 0.89 to 0.15, versus 0.88 to 0.50 for AgileThinker) This drop stems from its inability to consider future consequences of a move carefully. As exemplified in the case study in Figure 6, the reactive agent greedily pursues immediate rewards, falling into predictable traps while AgileThinker avoids by considering long-term survival requirements.

**Planning agent optimizes for decision quality but suffers under time pressure.** Planning agent excels under relaxed time constraints but suffers from dramatic degradation when time pressure increases (scores dropping from 0.92 to 0.05, versus 0.90 to 0.58 for AgileThinker). Its fundamental flaw is obliviousness to environmental changes, executing plans based on outdated observations. As illustrated in the case study, the agent is unaware that the snake has moved forward during reasoning, hence it crashes into a wall by following the obsolete plan. However, reactive thread in AgileThinker is informed of the latest state, thus able to adjust the output of planning thread.

The variants in §3 fail to balance speed and quality. R1 with budget forcing severely degrades performance ($0.01 < 0.39$ vs. V3), as forced truncation typically results in no-ops. Even advanced budget-aware methods (Figure 9) cannot prevent such drops. Meanwhile, R1 with code-policy succeeds only in simple algorithmic tasks like Freeway. It underperforms in complex scenarios where contextual reasoning (e.g., Theory-of-Mind) cannot be effectively compressed into code (see App. §C.4).

## 5 HOW TO MANAGE RESOURCES BETWEEN REACTION AND PLANNING?

Effective coordination between reactive thread $\mathcal{R}$ and planning thread $\mathcal{P}$ requires careful time management to determine when to invoke $\mathcal{R}$ within each environment step. We analyze how the token budget $N_{T_{\mathcal{R}}}$ allocated to $\mathcal{R}$ affects the performance of AgileThinker. We set $N_{T_{\mathcal{E}}}$ to 8k and vary the token budget $N_{T_{\mathcal{R}}}$ from 0.5k to 8k tokens. Figure 7 presents agent scores under different $N_{T_{\mathcal{R}}}$, where we also plot the cumulative distribution function (CDF) of $\mathcal{R}$'s token usage across all game trajectories without constraints to understand its inherent computational requirements.

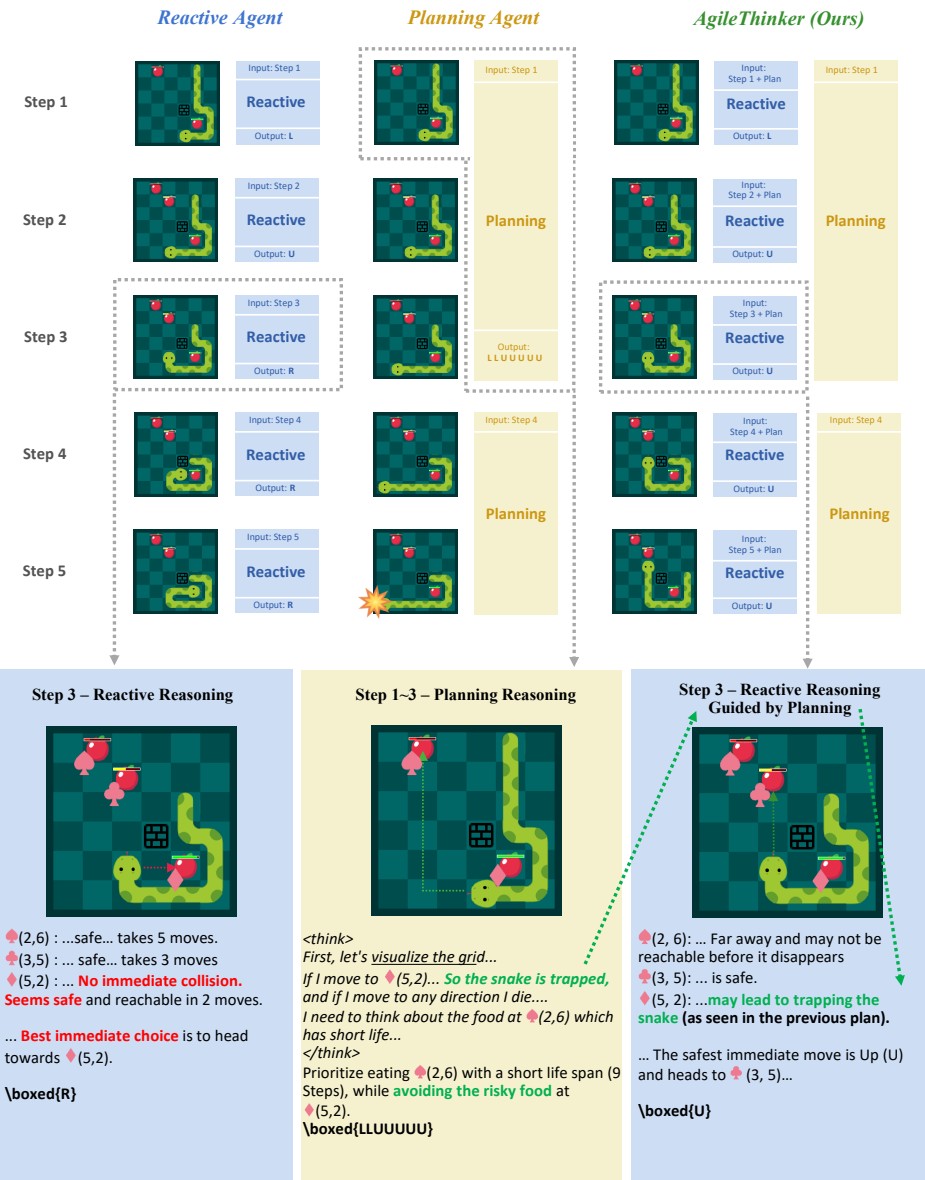

Figure 6: **Thinking trajectories of different paradigms at critical steps** At step 3, **Reactive Agent (V3)** greedily pursues the nearest food and collides inevitably after three steps. **Planning Agent (R1)**, still reasoning over the outdated step-1 state, defaults left. However, it correctly identifies that eating the nearest food would result in a future collision, and that its lifespan is sufficient to delay consumption. Guided by the reasoning of Reactive Thread, Planning Thread in the **AgileThinker** anticipates the trap and chooses to move upward toward a safer food target.

We can see that setting $N_{T_\mathcal{R}}$ too small (e.g., 0.5k) leads to low scores, as $\mathcal{R}$ doesn't have enough time to process strategic guidance from $\mathcal{P}$ and generate well-reasoned actions. Conversely, setting $N_{T_\mathcal{R}}$ too large creates idle periods where $\mathcal{R}$ has completed action generation but $\mathcal{P}$ continues productive reasoning. Empirically, performance peaks when $N_{T_\mathcal{R}}$ approximates the natural token upper bound of $\mathcal{R}$, as indicated by the CDF of $\mathcal{R}$'s token usage. This suggests that $\mathcal{R}$ benefits from fully utilizing its allocated time without truncation or extended idling.

It is worth noting that the optimal time budget varies across environments and requires empirical tuning. However, AgileThinker consistently outperform single-system baselines across broad budget ranges, suggesting that rough upper bound estimations are sufficient. In App.§ E, we further propose a dynamic adjustment mechanism for $N_{T_\mathcal{R}}$ at test time, which is also effective.

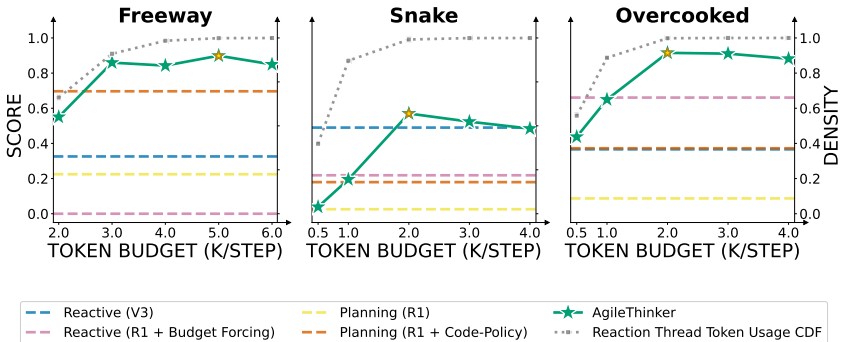

Figure 7: Performance of AgileThinker under different reactive thread token budgets $N_{T_{\mathcal{R}}}$. The cumulative distribution function (CDF) shows the natural token usage of $\mathcal{R}$ across all game trajectories when generation is not truncated, indicating inherent computational requirements of $\mathcal{R}$.

## 6 PERFORMANCE IMPROVEMENT UNDER WALL-CLOCK TIME

To validate the practical applicability of our token-based simulation, we conduct experiments using actual wall-clock time with official API of DeepSeek. Our results show token count has strong linear correlation with physical inference time. Specifically, we model this relationship as $T = \alpha N + \beta$, where

Table 2: Wall-clock time performance comparison across agent systems, confirming AgileThinker advantages persist in real-world deployment scenarios.

| Environment | Reactive (V3) | Planning (R1) | AgileThinker |
|---|---|---|---|
| Freeway | 0.24 | 0.12 | **0.88** |
| Snake | 0.37 | 0.04 | **0.45** |
| Overcooked | 0.57 | 0.00 | **0.89** |

$T$ represents total runtime and $N$ represents generated tokens. Least squares estimation on all experiment trajectories (plotted in Figure 10) yields $\alpha = 0.0473$ s/token, $\beta = 334.55$ s, with $R^2 = 0.9986$. This near-perfect correlation validates our token-based temporal abstraction and confirms its practical relevance for real-world deployments.

We also conduct experiments to verify that the advantage of AgileThinker remains when the game is simulated in wall-clock time. Using the derived TPOT of $0.047$ s/token, we evaluate agent systems with environment evolution intervals of $T_{\mathcal{E}} = 6$ minutes, corresponding to approximately 8,000 tokens per step. Table 2 shows that AgileThinker consistently outperforms both Reactive and Planning Agents in physical time. These results establish that our framework's benefits extend beyond theory to practical applications, demonstrating applicability of our architecture for agent deployments on intelligence-demanding, real-time tasks.

## 7 RELATED WORK

**Evaluation Environments for LLM Agents:** Existing evaluation setups for LLM agents mostly focus on *static* environments where nothing changes during episodes (Yang et al., 2024; Zhou et al., 2024b) or state pauses during LLM reasoning (Zhou et al., 2024a; Shi et al., 2025). This unrealistic assumption risks performance drop and even safety hazards when applying LLM agents in latency-sensitive applications (Sinha et al., 2024; Zheng et al., 2025). Prior work has modeled computation delays through Delay-Aware MDPs (Chen et al., 2020), sticky-action schemes (Mahmood et al., 2018), and asynchronous interactive MDPs (Travnik et al., 2018; Riemer et al., 2024), but the scope of these works is limited to traditional reinforcement learning. Although some works (Liu et al., 2024; Zhang et al., 2025) do adopt LLM agents in wall-clock time, our work is the first to formalize real-time reasoning problem for LLM Agents. In particular, we measure elapsed token count as a hardware-agnostic temporal unit, enabling fair and reproducible comparison across agent systems.

**Budget Control for Reasoning Models:** Test-time compute improves LLM performance but increases inference time, with overthinking behaviors commonly observed in current reasoning models (Chen et al., 2025). Budget control aims to maximize LLM performance under fixed budgets, and popular methods include early truncation (Muennighoff et al., 2025), prompting (Pu et al., 2025)

and training (Aggarwal & Welleck, 2025; Team et al., 2025; Gemini Team, 2025). These methods are effective to a certain extent, but still struggle with precise control over generated token count (Alomrani et al., 2025) and performance drops when budgets are far from adequate (Han et al., 2025). This suggests that existing techniques are inadequate for handling both loose and tight budget constraints within a single model. The results on Real-Time Reasoning Gym demonstrate that SOTA budget control methods cannot effectively balance reaction and planning, necessitating dual LLM architectures for real-time environments.

**Real-Time AI and Dual-Process Architectures:** The trade-off between accuracy and latency is a foundational problem in computer science, extensively studied ranging from real-time heuristic search to robotics (Kim et al., 2024). Classic approaches address this by dynamically allocating compute resources, such as using cascade classifiers for efficient detection (Chen & Yuille, 2005) or frame-skipping strategies in video tracking (Luo et al., 2018; Wang et al., 2020). While these paradigms are well-established in traditional AI, they are being revisited in the context of Large Language Models (LLMs), where variable inference costs introduce new challenges for real-time interaction (Ginart et al., 2024). Recent works have explored real-time capabilities in embodied agents (Kim et al., 2024) and asynchronous tool usage (Ginart et al., 2024), yet a unified architectural framework for balancing reasoning depth and reaction speed remains an open question.

To address this, researchers often draw inspiration from the dual process theory, which posits that human cognition operates through two distinct components: *System 1* (fast and intuitive) and *System 2* (slow and deliberate) (Evans, 2013; Kahneman, 2011). This cognitive model has motivated various dual-agent designs, combining fast modulessuch as finite-state machines (Zhang et al., 2025), vision transformers (Cui et al., 2025), or small language models (Liu et al., 2024)with powerful, computationally intensive modules like LLMs with tools (Christakopoulou et al., 2024) or Large Reasoning Models (LRMs) (Zhang et al., 2025). However, most existing architectures treat these systems as separate stages (e.g., cascading) or isolated parallel processes. AgileThinker distinctively advances this paradigm by allowing the *System 1* (Reactive Thread) to access the real-time partial reasoning traces of *System 2* (Planning Thread). This shared-context mechanism enables informed decision-making with minimal delay, effectively bridging the gap between classical real-time efficiency and modern LLM reasoning capabilities.

## 8 CONCLUSION

In this work, we identified and formalized **real-time reasoning**, a fundamental challenge faced by agents in real-world deployment. We introduced **Real-Time Reasoning Gym**, the first gym for evaluating LLM agents in continuously evolving environments. It supports independent control of cognitive load and time pressure, using token count as a hardware-independent temporal measure. Our evaluation revealed critical shortcomings of existing reasoning paradigms (reactive and planning reasoning). To address this gap, we proposed **AgileThinker**, which engages two reasoning paradigms in parallel. Experiments demonstrate that our method consistently outperforms all baselines, with advantages growing as cognitive load increases and time constraints tighten. Future work can extend our gym to more realistic scenarios, improve coordination mechanisms between two threads, or leverage our gym to train urgency-aware LLM agents.

## 9 LIMITATIONS

Our method investigates the real-time reasoning in LLM agents. Although our formulation is general, we only conducted experiments on DeepSeek models due to two reasons: (1) open source models perform poorly in general, thus the difference between different systems is not significant enough, and (2) other commercial model providers, including OpenAI, Google and Anthropic, do not provide reasoning traces which are crucial in our evaluation. We also try our best to make sure the readers understand that we do not have any empirical evidence showing that the dual system implemented in AgileThinker is precisely modeling human dual systems. The connection and difference require more rigorous evaluation.

## USE OF LANGUAGE MODELS

We used large language models to assist with paraphrasing and improving the readability of this paper. All research ideas, designs, experiments, analyses, and conclusions are entirely our own.

## REPRODUCIBILITY STATEMENT

We are committed to ensuring the reproducibility of our results. The implementation of Real-Time Reasoning Gym and AgileThinker, including all necessary code and dependencies, will be made publicly available on GitHub upon paper publication. The code can be run on any platform supporting DeepSeek V3 and R1 models, which we use for our experiments. Detailed instructions on how to set up and reproduce our results will be provided in the repository, including any additional dependencies and environment configurations.

## ACKNOWLEDGMENTS

We would like to thank reviewers, members of the SALT Lab, and the Stanford NLP Group for their feedback on this project. This work was supported by a grant under the NSF CAREER IIS-2247357 and ONR N00014-24-1-2532, as well as the Sloan Foundation.

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

## A    ENVIRONMENT DETAILS

- **Freeway:** The player navigates across parallel highways with moving cars. At each step the player can move to an adjacent lane or stay in place. Cars move forward at constant speeds and new cars may spawn on either side of the road. If the player is hit by a car, it will be reset to the starting position. The game terminates if the player crosses the road or if the step limit $M = 100$ is reached. The reward for a trajectory $\tau$ is computed as:

$$R(\tau) = M - |\tau|$$

- **Snake:** The player controls a snake in a 2D rectangular grid with surrounding walls and internal obstacles. At each step, the snake can move one step left, right, up or down. If the snake head collides with an obstacle or its body segment, it dies. Foods spawn continuously in the map and disappears after a fixed number of steps; eating food increases the snake's length by 1 unit. The game terminates if the snake dies or the number of steps exceeds threshold $M = 100$. The reward is calculated as:

$$R(\tau) = \text{Number of eaten food} - \mathbb{I}[\text{Dies in } M \text{ steps}]$$

- **Overcooked:** A fully observable two-player cooperative game where players must collect onions, cook them in a pot and serve the cooked soup for rewards. At each step, the players can move in 4 directions or use interact action to trigger some events, such as picking or placing an item depending on the game state. Since we focus on single-agent settings, we model the second player as part of the changing environment, controlled by a manually written script for simplicity and consistency. This agent randomly chooses one policy to follow: *deliver an onion into an arbitrary pot or a kitchen counter*. The game runs for $M = 100$ steps, and rewards are assigned for accomplishment of special events listed in Table 3. The game is implemented based on the repository `https://github.com/HumanCompatibleAI/overcooked_ai`

| Event | Reward |
|---|---|
| Picking up a dish from the dispenser | 3 |
| Picking up a cooked soup from the pot | 5 |
| Serving the soup | 20 |

Table 3: Rewards for different events in the Overcooked environment.

The reward is then normalized to $[0, 1]$ to get the final game score. Specifically, let $R_{\min}$ and $R_{\max}$ be the minimum and maximum rewards observed in all trajectories. The score $S$ is computed as: $S = \frac{R - R_{\min}}{R_{\max} - R_{\min}}$ Empirically, the $R_{\max}$ and $R_{\min}$ of each environment are listed in Table 4.

| Environment | $R_{\min}$ | $R_{\max}$ |
|---|---|---|
| Freeway | 0 | 89 |
| Snake | -1 | 15 |
| Overcooked | 0 | 56 |

Table 4: Minimum and maximum rewards for each environment.

Table 5: Game difficulty settings, showing ranges for easy, medium, and hard levels.

| Game | Dynamic Aspect | Cognitive Load Factor | Easy | Medium | Hard |
|---|---|---|---|---|---|
| Freeway | Hazards | Min steps to finish: $S$ | $S \leq 12$ | $13 \leq S \leq 16$ | $17 \leq S \leq 21$ |
| Snake | Opportunities | #Obstacles: $N$ | $N = 1$ | $2 \leq N \leq 5$ | $6 \leq N \leq 8$ |
| Overcooked | Partners | Kitchen Counter Len.: $L$ | $L = 0$ | $L = 3$ | $L = 4$ |

# B  PROMPT

**Prompt for the Planning Agent**

Now a player is playing a multi-turn game, and suppose current turn is $\{t_1\}$. Given the initial position $(0, y_{t_1})$ on a 2D grid (vertical axis $y = 0, 1, \ldots, 9$), determine the minimal number of turns $H$ and a sequence of actions $\{a_{t_1+t}\}_{t=0}^{H-1}$ to reach $(0, 9)$, avoiding collisions with cars on freeways $y = 1, \ldots, 8$.

---

**1. Game Dynamics:**

- **Player update:** $y_{t+1} = y_t + \Delta y_t$, where

$$\Delta y_t = \begin{cases} +1 & \text{if } a_t = U \\ -1 & \text{if } a_t = D , \quad y_{t+1} \in [0, 9] \\ 0 & \text{if } a_t = S \end{cases}$$

- **Car update rules:**
  For car $k$ on freeway $i$, suppose its head is at $h$, tail is at $\tau$ at turn $t_1$, and speed is $s$. Then at turn $T > t_1$, the car span becomes:

  - Left-moving: $\text{Span}(t_1) = [h, \tau] \rightarrow \text{Span}(T) = [h - s(T - t_1), \tau - s(T - t_1)]$
  - Right-moving: $\text{Span}(t_1) = [\tau, h] \rightarrow \text{Span}(T) = [\tau + s(T - t_1), h + s(T - t_1)]$
- Collision occurs at turn $T$ only if $0 \in \text{Span}(T)$ for any car on freeway $y_T$.

- Note that if you decide to move to $y_{T+1} \neq y_T$ at turn $T$, you will **NOT** be considered to be on $y_{T+1}$ at turn $T$, thus will **NOT** be collided by cars on $y_{T+1}$ if $0 \in \text{Span}(T)$ but $0 \notin \text{Span}(T+1)$.

---

**2. Task (Turn $t_1$):**

Find a sequence of actions $\{a_{t_1+t}\}_{t=1}^{H-1}$ which minimizes $H$ such that $y_{t_1+H-1} = 9$.

**Answer Format**:

Turn $t_1$: $a_{t_1}$
Turn $t_1 + 1$: $a_{t_1+1}$
$\vdots$

---

**3. Current State (Turn $t_1$):**

**Current Turn:** $t_0 = 10$

**Player Position:** $(0, 6)$

**Car State:**

| Freeway $k$ | Cars (head $h$, tail $\tau$, direction $d$, speed $s$) |
|---|---|
| 1 | $(48, 37, \text{right}, 12), (0, -11, \text{right}, 12)$ |
| 2 | $(48, 1, \text{right}, 48)$ |
| ... | ... |
| 8 | $(48, 37, \text{right}, 12), (0, -11, \text{right}, 12)$ |

**Prompt for the Reactive Agent**

---

**Prompt:**

You are a player in a freeway game, starting at $(0, y_{t_0})$ on a 2D grid (vertical axis $y = 0, 1, \ldots, 9$). Your goal is to reach $(0, 9)$ while avoiding collisions with cars on freeways $y = 1, \ldots, 8$.

---

**1. Game Dynamics:**

- **Player update:**
  $y_{t+1} = y_t + \Delta y_t$, where

$$\Delta y_t = \begin{cases} +1 & \text{if } a_t = U \\ -1 & \text{if } a_t = D \ , \quad y_{t+1} \in [0, 9] \\ 0 & \text{if } a_t = S \end{cases}$$

- **Car update rules:**
  For car $k$ on freeway $i$, suppose its head is at $h$, tail is at $\tau$ at turn $t_0$, and speed is $s$. Then at turn $T > t_0$, the car span becomes:

  - Left-moving: $\text{Span}(t_0) = [h, \tau] \rightarrow \text{Span}(T) = [h - s(T - t_0), \tau - s(T - t_0)]$
  - Right-moving: $\text{Span}(t_0) = [\tau, h] \rightarrow \text{Span}(T) = [\tau + s(T - t_0), h + s(T - t_0)]$

- Collision occurs at turn $T$ only if $0 \in \text{Span}(T)$ for any car on freeway $y_T$.

- Note that if you decide to move to $y_{T+1} \neq y_T$ at turn $T$, you will **NOT** be considered to be on $y_{T+1}$ at turn $T$, thus will **NOT** be collided by cars on $y_{T+1}$ if $0 \in \text{Span}(T)$ but $0 \notin \text{Span}(T+1)$.

---

**2. Guidance from a Previous Thinking Model (Turn $t_1 \leq t_0$):**

Sometimes, you have access to a past output from a thinking model, computed at turn $t_1$ based on then-current observations. This guidance may no longer perfectly match the current situation but can still be valuable for decision-making. You can use this plan as a **strategic reference**, not a mandatory instruction. Consider how much of the original strategy is still valid under the current dynamics.

---

**3. Task (Turn $t_0$):**

Choose **one** action $a_{t_0} \in \{U, D, S\}$ for the current turn, with the following considerations:

- **Collision Avoidance:** Ensure the action avoids both immediate and near-future collisions.
- **Strategic Consistency (Optional):** Refer to the thinking model's prior strategy. If the current environment still aligns with its assumptions, you may choose to continue along the same strategic direction. If not, adapt as needed.

**Answer Format:**

$$\boxed{a_{t_0}}$$

---

**4. Current State (Turn $t_0$):**

**Current Turn:** $t_0 = 10$

**Player Position:** $(0, 6)$

**Car State:**

| Freeway $k$ | Cars (head $h$, tail $\tau$, direction $d$, speed $s$) |
|---|---|
| 1 | $(48, 37, \text{right}, 12), (0, -11, \text{right}, 12)$ |
| 2 | $(48, 1, \text{right}, 48)$ |
| 3 | $(-48, -25, \text{left}, 24)$ |
| 4 | $(-48, -59, \text{right}, 12), (-12, -23, \text{right}, 12), (24, 13, \text{right}, 12)$ |
| 5 | $(-24, -35, \text{right}, 4)$ |
| 6 | $(-24, -35, \text{right}, 4), (52, 41, \text{right}, 4), (16, 5, \text{right}, 4)$ |
| 7 | $(48, 37, \text{right}, 12), (0, -11, \text{right}, 12)$ |
| 8 | $(48, 37, \text{right}, 12), (0, -11, \text{right}, 12)$ |

**Guidance from a Previous Thinking Model:** Turn $t_1 = 10$

Previous thinking process:

> First, I need to get the player from $(0, 6)$ to $(0, 9)$ in as few turns as possible, avoiding collisions with cars on the freeways $y = 1$ to $y = 8$. The current turn is $t_1 = 10$.
>
> The player can move up (U), down (D), or stay (S), changing $y$ by $+1$, $-1$, or $0$ respectively, and $y$ must stay between 0 and 9 inclusive.
>
> Cars move on specific freeways, and I need to avoid being on a freeway when a car is at position $x = 0$ at that turn. The collision happens only if at turn $T$, the player is on freeway $y_T$, and for that freeway, there's a car where 0 is in its span at that exact turn.
>
> Importantly, when I move to a new freeway, I'm not considered on that freeway until the next turn. So if I move from $y$ to $y + 1$ at turn $T$, at turn $T$ I'm still on $y$, and I only occupy $y + 1$ at turn $T + 1$. That means I don't collide with cars on $y + 1$ at turn $T$, only at turn $T + 1$ if I'm still there.
>
> Now, the goal is to reach $y = 9$. Since we start at $y = 6$, and $y = 9$ is three levels up, but I have to go through $y = 7$ and $y = 8$, and avoid the cars there.
>
> The minimal number of turns might be more than 3 because I might need to wait or something to avoid collisions.
>
> $H$ is the number of turns, so from turn $t_1$ to $t_1 + H - 1$, so if I start at $t = 10$, and I take $H$ turns, I end at turn $10 + H - 1$.
>
> I need $y_{10+H-1} = 9$.
>
> Since I start at $y = 6$, and I need to get to $y = 9$, which is $+3$, so minimum $H$ would be 3 if I can go up three times without colliding, but I need to check if that's possible given the car positions.
>
> But the cars are moving, so I need to see when they are at $x = 0$. First, let's understand the car movements.
>
> The car spans are given at turn t1=10.
>
> For each car, I need to find when 0 is in its span for future turns.
>
> The span changes linearly with time based on speed and direction.
>
> Let me denote the time elapsed from t1=10 as $\Delta t = T - 10$.
>
> For a car, at turn T, span is:
>
> If left-moving: [h - s*$\Delta t$, $\tau$ - s*$\Delta t$] since it moves left, decreasing x.
>
> In the rule: for left-moving, Span(T) = [h - s*(T-t1), $\tau$ - s*(T-t1)]
>
> Similarly for right-moving: Span(T) = [$\tau$ + s*(T-t1), h + s*(T-t1)]
>
> And collision if 0 is in Span(T) for the freeway we're on at turn T.
>
> Now, since the player is at y, and cars are per freeway, I need for each freeway k, and for each car on that freeway, find the times when 0 is in its span.
>
> Then, to avoid collision, when I'm on freeway k at turn T, none of the cars on k should have 0 in span at T.
>
> Now, for the path to y=9.
>
> Since y=9 is the top, and no cars above, but I need to pass through y=7 and y=8.
>
> y=9 has no cars since cars are on y=1 to 8, so once I reach y=9, I'm safe.
>
> But I need to get there.
>
> Start at y=6 at t=10.
>
> I need to go to y=7, then y=8, then y=9.
>
> But I might stay or move down to avoid cars, but moving down might not help since I need to go up eventually.
>
> Since the grid is small, y from 0 to 9, and I start at 6, going to 9, it's only 3 steps up, so H min is 3, but if not possible, I need more turns by staying or moving down and up.
>
> ... ...

**Prompt for Code-As-Policy**

**Prompt:**

Now a player is playing a multi-turn game, and suppose current turn is $\{t_1\}$. Given the initial position $(0, y_{t_1})$ on a 2D grid (vertical axis $y = 0, 1, \ldots, 9$). You are tasked with implementing a Python function that determines the optimal next action to avoid collisions with moving cars and progress towards the goal position $(0, 9)$ in minimal turns.

---

**1. Game Dynamics:**

- **Player update:**
  $y_{t+1} = y_t + \Delta y_t$, where

  $$\Delta y_t = \begin{cases} +1 & \text{if } a_t = U \\ -1 & \text{if } a_t = D \ , \quad y_{t+1} \in [0, 9] \\ 0 & \text{if } a_t = S \end{cases}$$

- **Car update rules:**
  For car $k$ on freeway $i$, suppose its head is at $h$, tail is at $\tau$ at turn $t_1$, and speed is $s$. Then at turn $T > t_1$, the car span becomes:

  - Left-moving: $\text{Span}(t_1) = [h, \tau] \rightarrow \text{Span}(T) = [h - s(T - t_1), \tau - s(T - t_1)]$
  - Right-moving: $\text{Span}(t_1) = [\tau, h] \rightarrow \text{Span}(T) = [\tau + s(T - t_1), h + s(T - t_1)]$

- Collision occurs at turn $T$ only if $0 \in \text{Span}(T)$ for any car on freeway $y_T$.

- Note that if you decide to move to $y_{T+1} \neq y_T$ at turn $T$, you will **NOT** be considered to be on $y_{T+1}$ at turn $T$, thus will **NOT** be collided by cars on $y_{T+1}$ if $0 \in \text{Span}(T)$ but $0 \notin \text{Span}(T + 1)$.

---

**2. Task**

You need to determine the best next action for the player by generating an **executable** Python function `next_action(json_state)` with the input `json_state` representing the current game state as a JSON object. The function should analyze the game state and return the next action, represented as a single character string:

- 'U' for moving up (to $y + 1$)
- 'D' for moving down (to $y - 1$)
- 'S' for staying in the current position

Notice that the code will be executed in a loop, so it should return the next action each time it is called with the current game state, which will **change after each action**.

**Input Format**

```
json_state = {
    'player_states': current_y_position,  # int: 0-9, 9 is the goal
        position
    'car_states': [  # list of tuples
        (lane, head, direction, speed, span),
        # lane: 1-8 (freeway number)
        # head: int, position of the car's head
        # direction: 'left' or 'right', tail = head + span if left,
            head - span if right
        # speed: int, speed of the car
        # span: int, span of the car, defined as the absolute
            difference between head and tail
    ],
    'turn': current_turn_number  # int: current turn
}
```

**Output Format**

Generate **two clearly labeled parts**:

**Part 1: Summary**

> One-sentence intent describing your strategy for the next actions

**Part 2: Python Function**

```python
def next_action(json_state) -> str:
    """
    Returns one of the actions: 'U', 'D', or 'S' based on the
        current game state.

    Args:
        json_state: The current game state as JSON object

    Returns:
        str: Single character representing the next action ('U', 'D
            ', 'S')
    """
    # Your logic here
    return action
```

**Example Output**

**Part 1: Summary**

> Wait for the cars in lane 4 and 5 to pass, then move up to avoid collisions.

**Part 2: Python Function**

```python
def next_action(json_state) -> str:
    # Implementation...
    return 'S'  # default action if no immediate threat
```

**Current State**

```
{
    "player_states": 5,
    "car_states": [
        (1, -39, 'right', 3, 11), (2, 48, 'right', 12, 11),
        (2, -36, 'right', 12, 11), (2, 0, 'right', 12, 11),
        (3, 48, 'right', 12, 11), (3, -36, 'right', 12, 11),
        (3, 0, 'right', 12, 11), (4, 48, 'right', 12, 11),
        (4, -36, 'right', 12, 11), (4, 0, 'right', 12, 11),
        (5, -18, 'right', 6, 11), (6, -18, 'right', 6, 11),
        (7, -16, 'right', 4, 11), (8, -16, 'right', 4, 11)
    ],
    "turn": 8
}
```

# C Additional Experiment Results

## C.1 Main Results

Table 6: Complete agent performance across various cognitive load levels (Easy, Medium, Hard) with time pressure fixed at 8k tokens/step.

**Freeway**

| Cognitive Load | Code-as-Policy | Reactive (R1) | Reactive (V3) | Planning (R1) | AgileThinker |
|---|---|---|---|---|---|
| Easy | 0.5393 | 0.2022 | 0.9775 | 0.3371 | 0.9551 |
| Medium | 0.6966 | 0.0000 | 0.3258 | 0.2247 | 0.8427 |
| Hard | 0.5281 | 0.0000 | 0.0562 | 0.1011 | 0.5056 |

**Snake**

| Cognitive Load | Code-as-Policy | Reactive (R1) | Reactive (V3) | Planning (R1) | AgileThinker |
|---|---|---|---|---|---|
| Easy | 0.1719 | 0.1719 | 0.7694 | 0.0588 | 0.6931 |
| Medium | 0.1797 | 0.2188 | 0.4900 | 0.0256 | 0.5413 |
| Hard | 0.0625 | 0.1406 | 0.2950 | 0.0137 | 0.3906 |

**Overcooked**

| Cognitive Load | Code-as-Policy | Reactive (R1) | Reactive (V3) | Planning (R1) | AgileThinker |
|---|---|---|---|---|---|
| Easy | 0.4621 | 0.8193 | 0.9188 | 0.5379 | 1.0000 |
| Medium | 0.3724 | 0.6607 | 0.3664 | 0.0871 | 0.9152 |
| Hard | 0.3661 | 0.2054 | 0.0877 | 0.0000 | 0.5982 |

Table 7: Complete agent performance across time pressure levels (4k to 32k tokens/step) with cognitive load fixed at Medium.

**Freeway**

| Tokens/Turn | Code-as-Policy | Reactive (R1) | Reactive (V3) | Planning (R1) | AgileThinker |
|---|---|---|---|---|---|
| 32k | 0.9438 | 0.0000 | 0.2911 | 0.9621 | 0.9431 |
| 16k | 0.9551 | 0.0000 | 0.2911 | 0.9045 | 0.9347 |
| 8k | 0.6966 | 0.0000 | 0.2911 | 0.2261 | 0.8469 |
| 4k | 0.3483 | 0.0000 | 0.2911 | 0.1194 | 0.6166 |

**Snake**

| Tokens/Turn | Code-as-Policy | Reactive (R1) | Reactive (V3) | Planning (R1) | AgileThinker |
|---|---|---|---|---|---|
| 32k | 0.2109 | 0.0238 | 0.4844 | 0.9629 | 0.8281 |
| 16k | 0.2344 | 0.0238 | 0.4844 | 0.4043 | 0.7813 |
| 8k | 0.1797 | 0.0238 | 0.4844 | 0.0254 | 0.5410 |
| 4k | 0.0156 | 0.0238 | 0.4844 | 0.0176 | 0.4238 |

**Overcooked**

| Tokens/Turn | Code-as-Policy | Reactive (R1) | Reactive (V3) | Planning (R1) | AgileThinker |
|---|---|---|---|---|---|
| 32k | 0.5804 | 0.0000 | 0.3800 | 0.8371 | 0.9129 |
| 16k | 0.4621 | 0.0000 | 0.3800 | 1.0000 | 0.9375 |
| 8k | 0.3661 | 0.0000 | 0.3800 | 0.0871 | 0.9152 |
| 4k | 0.3724 | 0.0000 | 0.3800 | 0.0246 | 0.7087 |

## C.2 SIGNIFICANCE TEST

We investigate the significance of advantage of AgileThinker over single-paradigm agents. We hypothesize that: (1) AgileThinker's advantage over reactive agents (V3) becomes more significant as cognitive load increases, and (2) its advantage over planning agents (R1) becomes more significant as time pressure increases.

To validate these hypotheses, we perform experiments across 3 cognitive load levels (Easy, Medium, Hard) and three 3 pressures (High: 32k tokens/step, Medium: 8k tokens/step, Low: 4k tokens/step). Below, we formally describe the validation procedure (1) and (2) can be tested similarly.

For each fixed cognitive load and time pressure condition, we analyze the paired score differences between AgileThinker and the reactive agent. Let $\mu_d$ denote the mean score difference between the paired observations (AgileThinker minus reactive agent). We pair observations that share the same environmental configuration seed, with each score calculated as the average across experimental runs within that configuration seed. We formulate the hypotheses as follows:

- **Null Hypothesis**: $\mathbf{H_0} : \mu_d = 0$

- **Alternative Hypothesis $\mathbf{H_1}$** : $\mu_d > 0$

The test statistic for the paired t-test is calculated as:

$$t = \frac{\bar{d}}{s_d/\sqrt{n}}$$

where $\bar{d}$ is the mean score difference, $s_d$ is the standard deviation of the differences, and $n$ is number of environment configuration seeds. We estimate the p-value based on the t-statistic with $n-1$ degrees of freedom.

Figure 8 presents the p-values across different conditions, with statistical significance assessed at $\alpha = 0.05$. The results show that AgileThinker's advantage generally becomes statistically significant as cognitive load and time pressure increase.

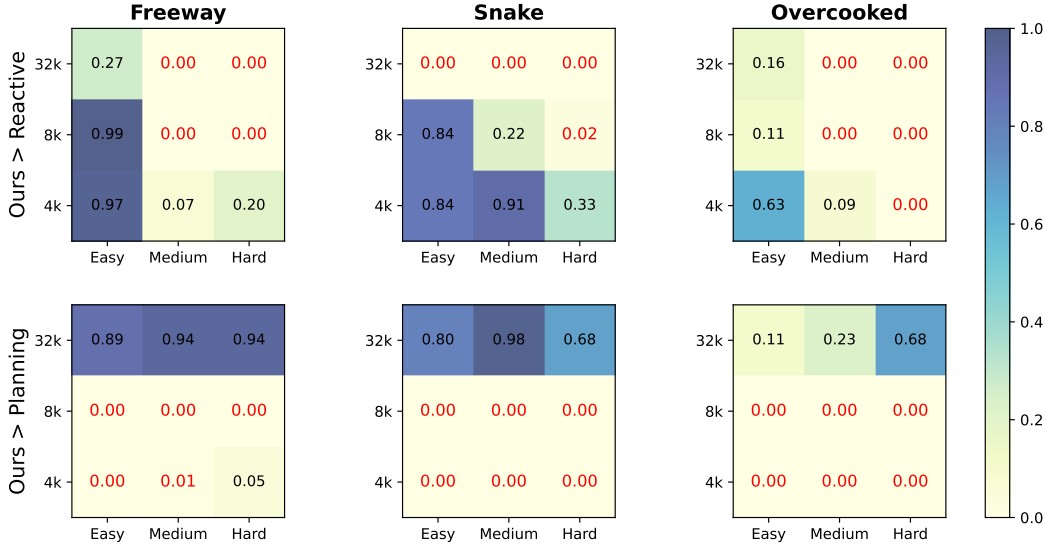

Figure 8: Statistical significance of AgileThinker's advantage over single-paradigm agents. Upper: Advantage over reactive agent (V3). Lower: Advantage over planning agent (R1). Numbers represent p-values under varying cognitive loads and time pressures, with red indicating statistical significance ($p < 0.05$). The advantage of AgileThinker generally increases with both cognitive load and time pressure.

Table 8: DeepSeek-V3.2 based agents' performance across different time pressure levels (4k to 32k tokens/step) with cognitive load fixed at High.

**Freeway**

| Tokens/Turn | Reactive (thinking off) | Planning (thinking on) | AgileThinker |
|---|---|---|---|
| 32k | 0.4659 | 0.8539 | 0.8889 |
| 16k | 0.4659 | 0.4649 | 0.8328 |
| 8k | 0.4659 | 0.1025 | 0.6352 |
| 4k | 0.4659 | 0.0000 | 0.4466 |

**Snake**

| Tokens/Turn | Reactive (thinking off) | Planning (thinking on) | AgileThinker |
|---|---|---|---|
| 32k | 0.2556 | 0.7381 | 0.5700 |
| 16k | 0.2556 | 0.3906 | 0.5350 |
| 8k | 0.2556 | 0.0550 | 0.3456 |
| 4k | 0.2556 | 0.0075 | 0.2738 |

**Overcooked**

| Tokens/Turn | Reactive (thinking off) | Planning (thinking on) | AgileThinker |
|---|---|---|---|
| 32k | 0.5246 | 0.8371 | 0.8550 |
| 16k | 0.5246 | 0.4554 | 0.8438 |
| 8k | 0.5246 | 0.0134 | 0.6563 |
| 4k | 0.5246 | 0.0068 | 0.4486 |

Table 9: DeepSeek-V3.2 based agents' performance with time pressure of 350 seconds per step and high cognitive load (Hard difficulty), confirming AgileThinker's advantage persists in wall-time constraints.

| Environment | Reactive (thinking off) | Planning (thinking on) | AgileThinker |
|---|---|---|---|
| Freeway | 0.3665 | 0.0 | 0.7022 |
| Snake | 0.3050 | 0.0863 | 0.6013 |
| Overcooked | 0.5246 | 0.0 | 0.6675 |

### C.3 RESULTS OF OTHER MODELS

Since AgileThinker relies on transparent reasoning trajectories, which are only available in open-source models, our primary experiments are conducted using the state-of-the-art open-source DeepSeek models. Besides using DeepSeek-V3 and DeepSeek-R1 as the reactive and planning model, we also conduct experiments with DeepSeek-V3.2, where non-thinking mode is used as reactive and thinking mode is used as planning. The results are shown in Tab. 8 and Tab. 9.

To assess the generalizability of our approach, we also evaluate reactive and planning agents using other models, such as Gemini-2.5-Flash, which features an intrinsic budget control function. For reactive agents, we disable extended thinking, while for planning agents, thinking is enabled.

However, we cannot directly implement AgileThinker with Gemini due to the lack of access to its reasoning traces. Instead, we approximate the system using a reduced design: the reactive agent references the *final* output of the planning agent after it completes reasoning, rather than accessing *partial* reasoning traces during streaming. Despite this limitation, the results in Tab. 10 consistently show that combining reactive and planning paradigms improves performance across all levels of time pressure.

Additionally, we observe that Gemini-2.5-Flash's built-in budget control struggles to precisely regulate token usage, often exceeding the allocated budget (Figure 9). This highlights the ongoing challenge that LLMs face in controlling computational costs during real-time reasoning.

Table 10: **Performance of Gemini-2.5-Flash (Medium-difficulty Freeway) under various agent designs.** We implement budget control by setting the thinking budget equal to the time pressure budgets. Since Gemini-2.5-Flash's internal reasoning traces are not public, we cannot apply Agile-Thinker directly. Instead, we reduce Reactive + Planning to allowing reactive thread to reference the *final* non-thinking tokens produced by planning thread after its reasoning is completed.

| Tokens/Step | Reactive (thinking off) | Reactive (thinking on + budget control) | Planning (thinking on) | Reactive+ Planning |
|---|---|---|---|---|
| 32k | 0.12 | 0.93 | 0.93 | 0.92 |
| 16k | 0.12 | 0.76 | 0.70 | 0.70 |
| 8k | 0.12 | 0.09 | 0.25 | 0.31 |
| 4k | 0.12 | 0.00 | 0.05 | 0.26 |

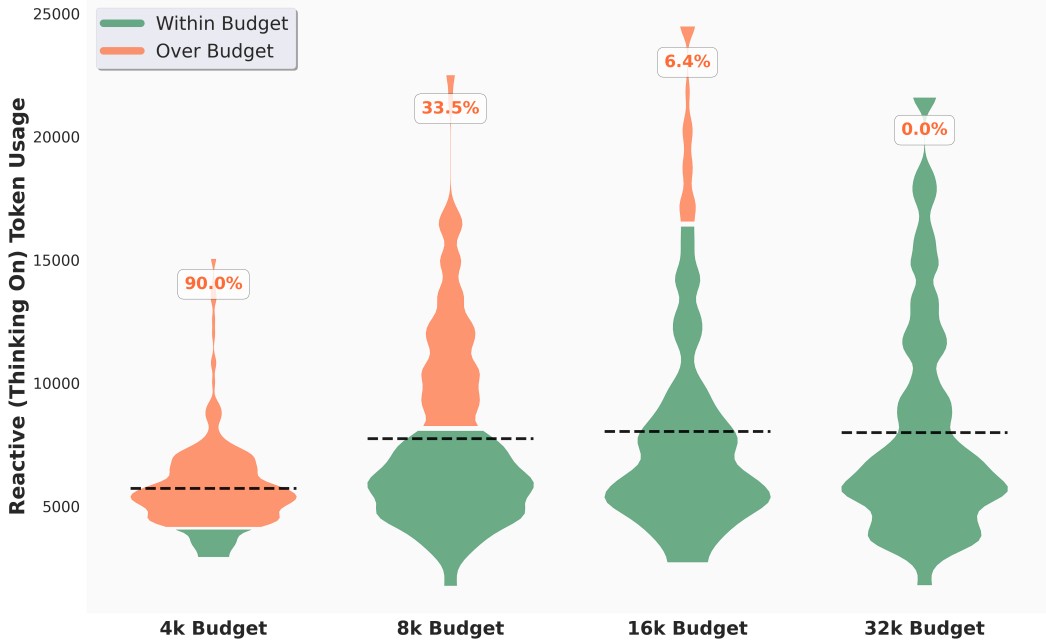

Figure 9: To test the effectiveness of thinking budget control, we set the thinking budget of Gemini-2.5-Flash at 4k, 8k, 16k and 32k to investigate the distribution of response length. The distribution shows that even Gemini-2.5-Flash is aware of the budget, it cannot precisely regulate response token count, often generating excessive tokens.

## C.4 CODE-AS-POLICY

We analyze generated code to understand the limitations of code-as-policy approaches. Below are representative examples from each environment. All code is directly generated by DeepSeek-R1 with some helper functions omitted for brevity.

FREEWAY

```
1    def next_action(json_state) -> str: # Extract game state # Helper
                                        functions
2    ... t0, y0, cars = extract_game_state(json_state) max_time = t0 + 50
                                        # Reasonable
```

```
3     time horizon visited = set() queue = deque() start = (t0, y0, []) # (
                                        current_turn,
4     y, path_actions) visited.add((t0, y0)) queue.append(start)
5
6     while queue: # BFS path finding ... # BFS found no path: choose
                                        safest
7     immediate next action ...
```

**Freeway** has a simple, efficient solution: breadth-first search (BFS) to explore all possible movement sequences across lanes, and the state space is very small (time × player position). Therefore, LLMs can implement naive search strategies to get optimal performance.

SNAKE

```
1     def next_action(json_state) -> str: # Extract game state # Helper
                                        functions
2     ... # Main logic: choose best move for current state safe_moves =
                                        generate_safe_moves(json_state)
3     if not safe_moves: return 'U' # no safe move
4
5     best_move = None best_score = -float('inf') depth = 5 # Limited depth
                                        search
6
7     for move in safe_moves: next_state, immediate_reward = simulate_move(
                                        json_state,
8     move) future_reward = search(next_state, depth - 1) total_reward =
                                        immediate_reward
9     + future_reward if total_reward > best_score: best_score =
                                        total_reward
10    best_move = move
11
12    return best_move
```

**Snake** does not have an algorithmic solution with bounded complexity, forcing LLMs to resort to heuristic approaches. The generated code typically implements limited-depth search (5 turns in this example) to evaluate immediate rewards, but this constrained horizon proves insufficient for complex scenarios. Long-term consequences - such as distant food opportunities or collision risks that emerge beyond the search depth - remain invisible to the policy.

OVERCOOKED

```
1     def next_action(json_state: dict) -> str: # Extract game state #
                                        Helper functions
2     ... alice_held_obj = alice.get('held_object')
3
4     # Decision logic if alice_held_obj and alice_held_obj.get('name') ==
                                        'onion':
5     # Find available pots and add ingredients ...
6
7     elif alice_held_obj and alice_held_obj.get('name') == 'plate': # Pick
                                        up
8     ready soup ... elif not alice_held_obj: # Priority 1: Get plate for
                                        ready
9     soups ... # Priority 2: Get ingredients for cooking ...
10
11    return 'S' # Stay as default
```

```
1     def next_action(json_state: dict) -> str: if alice_held_obj.get('name
                                        ') == 'onion':
2     pot = find_pot() add_ingredient(pot) elif alice_held_obj.get('name')
                                        == 'plate':
```

```
3    soup = find_ready_soup() pickup_ready_soup() elif not alice_held_obj:
                                            # Priority
4    1: Get plate for ready soups # Priority 2: Get ingredients for
                                            cooking ... return
5    'S' # Stay as default
```

**Overcooked** has complex game context, demanding that players infer partner intentions through action history and plan coordinated responses based on current state. However, the generated code considers only limited context (e.g., what Alice is currently holding). This narrow focus leads to suboptimal decisions. For instance, when the agent holds an onion but all pots are occupied (perhaps by Bob), it simply stays idle rather than dropping the onion to pick up a plate and serve ready soup.

```
1    def next_action(json_state: dict) -> str: # Extract game state #
                                            Helper functions
2    ...
3
4    pots = [tuple(pos) for pos in layout['Pot']] ready_soups = [obj for
                                            obj in objects
5    if obj.get('name') == 'soup' and obj.get('is_ready', False)]
                                            cooking_soups =
6    [obj for obj in objects if obj.get('name') == 'soup' and obj.get('
                                            is_cooking',
7    False)]
8
9    alice_held_obj = alice.get('held_object')
10
11   # Priority 1: Serve ready soup if ready_soups: if alice_held_obj and
                                            alice_held_obj.get('name')
12   == 'soup': # Deliver soup to serving counter ... elif alice_held_obj
                                            and
13   alice_held_obj.get('name') == 'plate': # Pick up ready soup from pot
                                            ... else:
14   # Get plate from dispenser ...
15
16   # Priority 2: Handle cooking soups elif cooking_soups: soup =
                                            cooking_soups[0]
17   if soup['remaining_cooking_tick'] <= 1 and not alice_held_obj: # Get
                                            plate for
18   soon-to-be-ready soup ... # Priority 3: Start cooking (add
                                            ingredients to empty
19   pot) idle_pot = next((pot for pot in pots if not any(obj.get('
                                            position') == list(pot)
20   for obj in objects)), None) if idle_pot: if held_obj and held_obj.get
                                            ('name')
21   == 'onion': # Add ingredient to pot ... else: # Get ingredient from
                                            dispenser
22   ...
23
24   return 'S' # Stay as default
```

Beyond ignoring broad game context, the generated code exhibits inconsistent goal prioritization that leads to oscillatory behavior. Consider a scenario where Alice holds an onion and ready soup is available: the first code prioritizes adding the onion to an empty pot, while the second prioritizes serving the ready soup. These conflicting objectives cause the agent to alternate between incompatible actions. This demonstrates a fundamental limitation of code-as-policy approaches: while they can encode local heuristics effectively, they struggle to maintain coherent long-term strategies in environments requiring dynamic coordination and context-aware decision making.

## C.5 PERFORMANCE UNDER LIMITED THROUGHPUT

| Game | Parallel Threads | Concurrent Threads | Reactive (V3) | Planning (R1) |
|---|---|---|---|---|
| Freeway | 0.84 | 0.72 | 0.32 | 0.22 |
| Snake | 0.54 | 0.58 | 0.49 | 0.03 |
| Overcooked | 0.92 | 0.85 | 0.37 | 0.09 |

Table 11: Performance of AgileThinker with abundant (parallel threads) and limited (concurrent threads) throughput.

While our main experiments assume parallel execution with independent throughput for each subsystem, practical deployments may face resource constraints. In this section, we investigate whether AgileThinker remains effective when LLM and LRM share computational resources through concurrent switching rather than parallelism. Specifically, we evaluate the performance of agent systems under equivalent token throughput, implementing the AgileThinker as alternating inference between reactive and planning thread (concurrent generation) rather than simultaneous generation.

Figure 11 compares concurrent AgileThinker against parallel AgileThinker and single-model baselines. Even with equivalent throughput, concurrent AgileThinker significantly outperform both reactive agents and planning agents. While parallel execution offers modest performance improvements over concurrent execution, the gains are relatively small compared to the fundamental advantage of AgileThinker. This indicates that the primary benefit stems from cognitive specialization rather than computational resources, and AgileThinker remain effective for resource-constrained deployments.

## D WALLTIME EXPERIMENTS

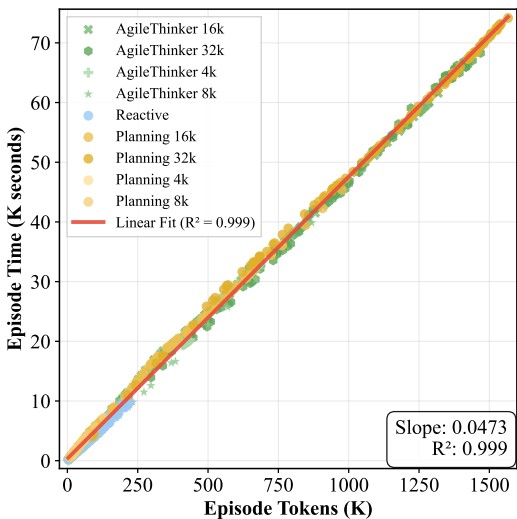

Figure 10: Almost linear correlation between generated token count and wall-clock time using DeepSeek official API, demonstrating the validity of our token-as-time abstraction. Here the numbers after agent methods, e.g. 4k, 8k, refer to the corresponding environment time pressure budgets.

## E DYNAMIC ADJUSTMENT ALGORITHM FOR INTERNAL BUDGET

Inspired by the AIMD algorithm [2], we conduct additional experiments to show that thread resource can be allocated adaptively through a simple dynamic adjustment mechanism for $N_{T_{\mathcal{R}}}$. Specifically,

---
[2] https://en.wikipedia.org/wiki/Additive_increase/multiplicative_decrease

we initialize $N_{T_{\mathcal{R}}}$ to a small value (1k tokens). If the reactive thread fails to generate a complete response within the allocated tokens, we increase $N_{T_{\mathcal{R}}}$ by 1k tokens. Conversely, if the reactive thread uses fewer than tokens for 3 consecutive responses, we decrease $N_{T_{\mathcal{R}}}$ by 0.5k tokens. This approach eliminates the need to predefine a fixed $N_{T_{\mathcal{R}}}$, allowing the model to adapt it based on query complexity.

In fact, the advantage of AgileThinker is still significant with this adaptive algorithm, as we can see in Tab. 12. The experiments are conducted on DeepSeek-V3.2 under **Hard** cognitive load and **8k tokens/step** time pressure.

---

**Algorithm 1** Dynamic Budget Adjustment

---

1: $N_{T_{\mathcal{R}}} \leftarrow 1024$, efficient_streak $\leftarrow 0$
2: **while** game not terminates **do**
3:     Execute AgileThinker with internal budget $N_{T_{\mathcal{R}}}$
4:     **if** reactive thread exceeds budget **then**
5:         $N_{T_{\mathcal{R}}} \leftarrow N_{T_{\mathcal{R}}} + 1024$
6:         efficient_streak $\leftarrow 0$
7:     **else**
8:         efficient_streak $\leftarrow$ efficient_streak + 1
9:         **if** efficient_streak $\geq 3$ **then**
10:           $N_{T_{\mathcal{R}}} \leftarrow \max(1024, N_{T_{\mathcal{R}}} - 512)$
11:           efficient_streak $\leftarrow 0$
12:         **end if**
13:     **end if**
14: **end while**

---

| Environment | Reactive (Non-thinking) | Planning (Thinking) | AgileThinker (fixed, optimal $N_{T_{\mathcal{R}}}$) | AgileThinker (dynamic $N_{T_{\mathcal{R}}}$) |
|---|---|---|---|---|
| Freeway | 0.4659 | 0.1025 | 0.6352 (3k) | 0.6279 |
| Snake | 0.2556 | 0.0550 | 0.3456 (1k) | 0.4166 |
| Overcooked | 0.5246 | 0.0134 | 0.6563 (2k) | 0.6741 |

Table 12: Scores of AgileThinker under dynamic internal budget.

