# OpenReview forum: "Real-Time Reasoning Agents in Evolving Environments"
_ICLR.cc/2026/Conference — ICLR 2026 Poster_

### Official Review · Reviewer_EhbH · 2025-10-31

**Soundness:** 3
**Presentation:** 3
**Contribution:** 3
**Rating:** 8
**Confidence:** 3

**Summary:**

The paper introduces AgileThinker, a dual-thread architecture for LLM agents that must act while reasoning in dynamic environments. It defines the real-time reasoning problem, where environments are dynamic and the agent’s computation time directly affects outcomes. Using the new Real-Time Reasoning Gym benchmark, the authors show that combining a fast reactive thread (for immediate responses) with a slower planning thread (for long-term reasoning) enables a better balance between speed and deliberation. Experiments demonstrate that AgileThinker outperforms purely reactive or planning agents under varying cognitive load and time pressure, with results validated both in simulated and real wall-clock settings.

**Strengths:**

1) The paper introduces real-time reasoning as a new challenge for LLM agents and proposes AgileThinker, a dual-thread framework that combines fast, reactive behaviour with slow planning. This is a clear conceptual advance beyond existing static or turn-based agent paradigms.
2) The authors validate their approach through a carefully designed benchmark (Real-Time Reasoning Gym), multi-game testing, controlled ablations, and statistical significance tests (appendix). The experimental evidence is strong and aligns with the theoretical motivation.
3) The paper is well-written, well-illustrated, and accessible, effectively communicating complex timing and coordination concepts. I found the work really interesting.

**Weaknesses:**

1) All experiments use simulated game-like settings (Freeway, Snake, Overcooked); they may not fully represent complex real-world tasks (e.g., robotics, human-AI interaction). The results demonstrate concept validity but not necessarily scalability to multimodal or continuous-control domains.
2) The paper needs to explore comparable baselines like:
   - Li, Yaoru, Shunyu Liu, Tongya Zheng, and Mingli Song. 2025. “Parallelized Planning-Acting for Efficient LLM-Based Multi-Agent Systems.” arXiv [Cs.AI]. arXiv. http://arxiv.org/abs/2503.03505.

**Questions:**

1) What are some alternatives to TR, i.e., alternatives to fine-tuning this hyperparameter for each domain/task?
2) What are the foreseeable computational and latency overheads of running two LLM threads in real-world systems, especially under networked or resource-constrained deployment?
3) How does AgileThinker compare with Yaoru et al.? Unless there are strong reasons not to, this is a good baseline to test AgileThinker.

---

> ### Author Response · Authors · 2025-11-20
>
> Thank you for your valuable feedback and thoughtful comments! We have addressed your points below.
>
> **W1**:
>
> We appreciate your concern regarding real-world applicability. We believe our conclusions **generalize to real-world deployment** since the settings we choose **cover a wide variety of cognitive loads, time pressure, and even different kinds of real-time dynamics** (hazards, opportunities, partners). This comprehensiveness gives a full-picture of the challenges of real-time reasoning and the advantage of AgileThinker.
>
> Moreover, we choose to use game environments for following reasons:
>
> (1) They provide a **controlled** setting where we can systematically vary difficulty and urgency, allowing us to isolate and study the effects of our architectural design. Games have long served as testbeds for novel AI problems,  with solutions later applied to application-driven tasks (e.g., interactive NLP was studied in text games, and RL  was studied on Atari games).
>
> (2) **Real-world complexity is multi-dimensional**: In this paper, we primarily study the complexity of dynamics (static/dynamic). Other dimensions, such as modalities (text/vision/audio) and physical interactions (HAI/robotics control), are beyond our current scope. However, we believe that as LLM reliability and inference speed improve, future work can explore applying AgileThinker to multi-modal and continuous control tasks in more realistic settings.
>
> **W2**:
>
> (1) The paper you mentioned falls into the category of planning agents according to our taxonomy. Its core limitation is that the plan generated by its "plan thread" cannot take effect until it is finalized. In our architecture, the reactive thread can immediately use the partial output of the plan thread and take proactive actions.
>
> (2) The problem setting of the paper is different from ours. They try to reduce inference time in sequential, synchronous environment, while our focus is to get robust performance under time pressure.
>
>  We test its performance in our benchmark under **Hard** cognitive load, **8k tokens/step** time pressure, and the result is similar to planning agent. The results are obtained on DeepSeek-V3.2 (its full performance is in revision)
>
> |  | Reactive | Planning | AgileThinker | Parallelized Planning-Acting[1] |
> |--------|---------|----------|---------------------|--------------------------|
> | Freeway | 0.4659|0.1025|0.6352 | 0.2022 |
> | Snake | 0.2556 | 0.0550 | 0.3456 | 0.0083 |
> | Overcooked | 0.5246 | 0.0134 | 0.6563 | 0.0115 |
>
> - [1] Li, Yaoru, Shunyu Liu, Tongya Zheng, and Mingli Song. 2025. “Parallelized Planning-Acting for Efficient LLM-Based Multi-Agent Systems.” arXiv [Cs.AI]. arXiv. http://arxiv.org/abs/2503.03505.
>
> **Q1**:
>
> We developed an alternative approach inspired by the AIMD algorithm. We implemented a **dynamic adjustment mechanism** for $N_{T_\mathcal{R}}$.
> Specifically, we initialize $N_{T_\mathcal{R}}$ to a small value (1k tokens). If the reactive thread fails to generate a complete response within the allocated tokens, we increase $N_{T_\mathcal{R}}$ by 1k tokens. Conversely, if the reactive thread uses fewer than $N_{T_\mathcal{R}}$ tokens for 3 consecutive responses, we decrease $N_{T_\mathcal{R}}$ by 0.5k tokens. This eliminates the need for a fixed $N_{T_\mathcal{R}}$, allowing the model to adapt it based on query complexity.
>
> This method turns out to be effective. The scores under **Hard** cognitive load and **8k tokens/step** time pressure are shown below:
>
> |  | Reactive | Planning | AgileThinker (fixed, optimal $N_{T_\mathcal{R}}$) | AgileThinker (adaptive $N_{T_\mathcal{R}}$) |
> |--------|---------|----------|---------------------|--------------------------|
> | Freeway | 0.4659|0.1025|0.6352 (3k)| 0.6279 |
> | Snake | 0.2556 | 0.0550 | 0.3456 (1k) | 0.4166 |
> | Overcooked | 0.5246 | 0.0134 | 0.6563 (2k)| 0.6741 |
>
> **Q2**:
>
> (1) **Computation Overhead**: We discuss this question in Appendix C.5. In the main text, our implementation requires parallel computation when the reactive thread starts. There is a pre-filling computation overhead for the reactive thread, but this is negligible compared to the total generation time of LLMs.
>
> In Appendix C.5, we also run an experiment where token throughput is limited. In this scenario, we run the two threads concurrently instead of in parallel (i.e., the planning thread pauses when the reactive thread is active). The results show that AgileThinker still outperforms the baselines by a large margin. This indicates that the primary benefit stems from **cognitive specialization** rather than increased **computational resources**, and that AgileThinker remains effective for resource-constrained deployments.
>
> (2) **Network Latency (API-based evaluation)**: Assuming a vocabulary size $V$, the information passing rate is $\log V$ bits per token. To support 8k tokens/step at 1 step/s, we only need a bandwidth of approximately 16 KB/s, which is negligible for most modern networks.
>
> **Q3**: See **W2**.

---

> > ### Comment · Reviewer_EhbH · 2025-11-28
> >
> > Thank you for your responses. I found the work interesting and maintained my score.

---

### Official Review · Reviewer_sdJh · 2025-10-31

**Soundness:** 4
**Presentation:** 4
**Contribution:** 4
**Rating:** 8
**Confidence:** 4

**Summary:**

The authors propose Real-Time Reasoning Gym, an environment consisting of three games (Freeway, Snake, and Overcooked) that are designed to test LLM agents in dynamic environments. Dynamic environments are those where the environment progresses at a fixed rate regardless of whether the agent has been able to come up with an action or not. To evaluate how current methods perform in this new setting, the authors study both reactive agents (non-thinking models, budget forcing) as well as planning agents (thinking models, code as policies). They find that each type of agent has its limitations, motivating the novel AgileThinker agent which runs two parallel threads (one for reactive, one for thinking) and mostly matches or outperforms all baselines in all settings.

**Strengths:**

1. The work provides a very novel and interesting problem setting for language agents, lifting the unrealistic assumption that the environment waits for agents to execute their actions.
2. The motivation is clear and overall the paper is well-written
3. The quantitative results are very strong, and the authors additionally provide a qualitative case study further illustrating the benefits of AgileThinker.

**Weaknesses:**

I could not discern any significant weaknesses, but I do have some questions in a few places (see questions section).

**Questions:**

1. Figure 3: what does “reference” mean? And why is there two frames marked as “Reactive Agent Output” under the Static setting? I thought the agent only outputs actions, but here somehow the observations seem to be marked as output?
2. Page 4: “For each game, we normalize the scores by the highest score the agent could get in that game, so we always have a score between 0 and 1.” → could the authors clarify exactly how this normalization works maybe by giving an example?
3. Page 5: “(2) code plans where a thinking model is used to generate a code snippet that automatically produces actions based on observation input (Liang et al., 2022; Zhang et al., 2025). Although (1) is often easier to generate, (2) is more adaptive to potential changes.” → could the authors explain this a bit more? As in, why is generating code plans more adaptive to potential changes? Also, it’s not immediately obvious to me that generating code should fall under planning?
4. Figure 6: for the “Planning Reasoning” yellow box in the middle, why is there only two apples instead of 3? In all other cases there seems to be three.
5. Figure 6: is this a case study of real trajectories?
6. Figure 5: it seems that in the bottom right plot, AgileThinker does better than the planning methods even though the time pressure is very lenient. I would have expected these methods to be at best on par here, but somehow AgileThinker outperforms the planning methods - could the authors comment on why this might be the case?
7. Page 8: “we also plot the cumulative distribution function (CDF) of R’s token usage across all game trajectories without constraints” → does this mean for the purpose of creating the CDF, R gets to see all of P? Or how is the partial reasoning reference point determined if there is no constraints?
8. Table 2: what’s the cognitive load setting for these experiments?

---

> ### Author Response · Authors · 2025-11-20
>
> We sincerely appreciate your encouraging review! We are glad that you found our idea novel, our motivation  well-presented, and our results compelling.
>
> **For Question 1**: "Reference" indicates the inputs provided to the reactive agent from the planning agent's completed reasoning. The reactive agent also performs reasoning, so its output includes both the reasoning process and  the final action. We will clarify the figure labeling in our revision.
>
> **For Question 2**: We divide each agent's score by the maximum possible score for that game. Please see Appendix  A for detailed calculation.
>
> **For Question 3**: Code plans contain conditional logic (e.g., if-else statements) that handle multiple scenarios.  When states change, code generated from outdated observations can often still apply, whereas fixed action sequences cannot. We categorize code generation as planning because it involves extended reasoning to create a  general policy rather than selecting immediate actions.
>
> **For Question 4**: This illustrates our key point. The planning agent reasons on the frozen state from step 1 (three  apples), but by this frame, the environment has evolved and one apple has been consumed (two apples remain).  This demonstrates how planning agents work with outdated observations in dynamic environments.
>
> **For Question 5**: Yes. This is a case study of real trajectories.
>
> **For Question 6**: This is likely due to variance. We expect planning methods to perform at least as well as Agile Thinker under lenient time pressure, and the observed difference is small.
>
> **For Question 7**: Please refer to the explanation in our response to Reviewer TV6c's Weakness 1.
>
> **For Question 8**: Medium cognitive load, as it best reveals performance differences between methods across varying time pressures.

---

> > ### Comment · Reviewer_sdJh · 2025-11-26
> >
> > I thank the authors for answering all my questions. I maintain positive about the work and hence maintain my score.

---

### Official Review · Reviewer_hZPr · 2025-10-31

**Soundness:** 3
**Presentation:** 3
**Contribution:** 3
**Rating:** 6
**Confidence:** 4

**Summary:**

This paper introduces the real-time decision-making into LLM based agents. Instead of having an agent-environment loop, the proposed real-time reasoning gym imposes a time pressure on agent reasoning manifested in the form of token limits, and tests how well agents can complete tasks under the pressure of bounded reasoning. The proposed gym features three diverse synthetic agent environments with different challenges that arise from dynamic environments, and experiments show that previous LLM agents perform poorly on these challenges, especially when time-pressure is high, sheding light on a new direction of research.

**Strengths:**

1. Real-time reasoning of AI agents under time pressure is an under-evaluated direction in the current wave of AI agents research, and this paper presents a good setting and three diverse synthetic environments to prototype research in this direction. Experiments show that with increased time pressure and task complexity, existing agent performance decrease significantly from the case where the environment waits indefinitely for agent reasoning.

2. The paper presents AgileThinker, an agent design that combines a planner and a reactive agent in a manner that controls planner output with a token budget for the agent to stay under the overall allotted time pressure limit. This agent design, albeit simple, shows gains over purely planning or reactive agents especially when high time pressure is applied on the agent.

**Weaknesses:**

1. The assumption that token limit is a good proxy for actual wallclock time is valid only when all agents are implemented with the same LLM served with the same hardware. Even if hardware independence is desirable, it is unlikely that different models will share the same slope (TPOT) and intercept (whatever factor that goes into time-to-first-token), because of model size, model architecture, etc. The paper's formulation and experiments neglected this important consideration.

2. As also seen in the result in Section 6, one crucial factor the token budget limitation neglects is the time-to-first-token cost of LLM calls in these language agents. Contrary to what the authors lead the reader to believe, the step time in the results presented in the paper for these simulated games is not 100s of milliseconds (as those familiar with these games would believe), but minutes (5.5 minutes to be precise). This is an important caveat, and undermines the real-time nature of the gym itself, since the environment is still unduely waiting for the agent to perform indefinite internal operations.

3. The paper could do a better job contextualizing itself in the literature.

    1. There is a limited set of agent designs that this paper experimented with, especially open-source implementations of agents from prior work to help the reader understand whether the baselines used in this paper are valid or competitive, and more importantly, whether the proposed AgileThinker framework can be a drop-in improvement for existing agent implementations.
    2. Real-time agents have long been a subject of AI research, from best-first search (A* and variants) and robotics [1], to more recently real-time customer support AI agents [2]. The paper could use a better review of related work in this direction, and better distinguish the key contributions it's making. The concept might not be entirely new, but it's useful to call it out in the new paradigm of AI agent design and evaluation.

[1] OpenVLA: An Open-Source Vision-Language-Action Model. (https://arxiv.org/pdf/2406.09246)

[2] Asynchronous Tool Usage for Real-Time Agents (https://arxiv.org/pdf/2410.21620)

**Questions:**

1. Table 2 and Section 6: what's the metric shown here, and what's the wallclock limit for these agents?
2. How does the proposed framework generalize to cases where the environment updates at a cadence that is not predetermined, e.g., one update takes 1s and another takes 10s?
3. How does AgileThinker generalize to other agent designs?

---

> ### Author Response · Authors · 2025-11-20
>
> We appreciate your detailed and encouraging feedback. We are pleased that you consider our research question  worthy of study and our method concise and efficient.
>
> > **Weaknesses 1**: Token-as-proxy ignores hard-ware serving issues and different TPOT across different models.
>
> Thank you for raising this concern. We would like to direct your attention to Section 6, where we conduct experiments using wall-clock time, which naturally accounts for the hardware and model-specific factors you mentioned. In practice, these implementation details have minimal impact on overall performance given the substantial computational cost of generating reasoning trajectories.
> We note that our work focuses primarily on high-level agent system design rather than low-level system optimization. Our goal is to demonstrate the efficiency and necessity of parallel reasoning architectures in real-time  environments. To isolate these architectural benefits and control for confounding variables, we deliberately use  models from the same family (DeepSeek) throughout our experiments.
>
> > **Weaknesses 2**: The actual agent response time is minutes rather than milliseconds, undermining the real time nature of the benchmark.
>
> Thank you for this important observation. We acknowledge that current response times are indeed limited by existing API infrastructure and hardware constraints. The API latencies (**TPOT = 47ms/token**) we observe reflect the current state of publicly hosted services (e.g., DeepSeek and similar providers), under which it is impossible to carry out experiments in millisecond-level. However, we expect these limitations to be rapidly  addressed through ongoing improvements in: (1) more efficient small language models, (2) scaled inference in infrastructure, (3) advanced inference algorithms, (4) hardware acceleration, and (5) network optimization.
>
> Despite current infrastructure limitations, our benchmark is designed to reveal meaningful insights across varying levels of time pressure and cognitive load. Importantly, the ~5.5 minute response time still enables us to meaningfully distinguish between Reactive and Planning reasoning modes, which is central to our contribution. As in reference technology advances toward sub-second latency, our architectural findings and the parallel reasoning paradigm we propose will become increasingly applicable to truly real-time scenarios.
>
> > **Weaknesses 3-1**: The paper lacks comparisons with existing open-source agent implementations to validate  baseline competitiveness and demonstrate AgileThinker's generalizability.
>
> We acknowledge that testing additional open-source agent implementations would be valuable. However, conducting experiments in our multi-turn game settings with extended reasoning across 3 difficulty levels and 4 time constraints incurs substantial computational costs—approximately thousands of dollars per agent configuration.
>
> Given these constraints, we adopted a principled approach: rather than exhaustively testing all existing open source agents, we systematically evaluated representative high-level architectural paradigms (e.g., code-as-policy,  budget forcing) using controlled base models. Our design choices aim to capture the fundamental reasoning  strategies underlying most current open-source agents, allowing us to draw generalizable conclusions about architectural principles rather than implementation-specific details. For example, the paper mentioned by Reviewer EhbH falls in our implementation of planning agents with similar performance.
>
> || Reactive | Planning | AgileThinker | Multi-agent Planing-Acting[1]|
> |:-:|:-:|:-:|:-:|:-:|
> |Freeway|0.4659|0.1025|0.6352|0.2022|
> |Snake|0.2556|0.0550|0.3456|0.0083|
> |Overcooked|0.5246|0.0134|0.6563|0.0115|
>
> [1]Li, Yaoru, Shunyu Liu, Tongya Zheng, and Mingli Song. 2025. “Parallelized Planning-Acting for Efficient LLM-Based Multi-Agent Systems.” arXiv [Cs.AI]. arXiv. http://arxiv.org/abs/2503.03505.

---

> > ### Author Response · Authors · 2025-11-20
> >
> > > **Weaknesses 3-2**: While real-time agents have been studied before, the paper should better distinguish its  contributions within the modern LLM-based agent paradigm.
> >
> > We agree that real-time agents represent a long-standing topic in AI research, and we appreciate the references to  prior work. The cited works [1, 2]  do fit in the frame of our related work (Dual Agent Systems Section). We will update our related work to include them in our revision.
> >
> > Besides, we would like to clarify that our work mainly focuses on the LLM reasoning dimension of the real-time challenge.  The cited works primarily focus on the *event-driven* aspect, or managing asynchronous events and prioritizing actions in response to external stimuli. In contrast, our work examines real-time constraints from the *computational latency* perspective, specifically addressing how reasoning agents in the LLM era should balance extended  thinking, which introduces non-negligible latency, against timely responses.
> >
> > We note that defining "critical events" requiring interruption is non-trivial. For instance, in our gym environments, all state updates are equally important. If the agent were interrupted at every update, it would effectively  operate in purely reactive mode, which is one of the paradigms we study. Our contribution lies in exploring how  agents can simultaneously engage both reactive and planning reasoning to handle continuous environmental  changes while maintaining reasoning depth.
> >
> > We believe we have clarified our choice of token-as-time in W1, the reason for choosing a large period time in W2 and our unique contribution in W3 with additional review of related work. We would be very grateful if you could check these and reconsider your score if you think they have addressed your concerns.
> >
> > **For Question 1**: The metric reported in Table 2 is consistent with the definitions used in our main experiments. Regarding the time constraints, the wallclock limit per episode is set to $T_\mathcal{E} = 6$ minutes, as detailed in Section 6.
> >
> > **For Question 2**: Our current framework employs discrete time control, a standard approach for approximating continuous-time MDPs. To handle variable environmental update cadences, the framework can operate on a fixed base frequency where intervals without environmental changes are treated as "null" or "no-op" updates. Furthermore, as discussed in our response to Reviewer TV6c (Weakness 1), the framework can be extended with dynamic scheduling algorithms to adaptively synchronize with irregular environmental latencies without structural changes.
> >
> > **For Question 3**: AgileThinker is designed as a generalizable paradigm for real-time interaction. Its core architectural distinction lies in decoupling reasoning from execution via *parallel planning and reactive reasoning*. Consequently, this design is not limited to the specific underlying model; any agent operating in a real-time environment can adopt this dual-process structure to enhance responsiveness while maintaining long-horizon planning capabilities.
> >
> > We hope these responses address your concerns, and we are happy to provide further clarification if needed.

---

> ### Comment · Reviewer_hZPr · 2025-11-27
> **Thank you for the response.**
>
> Thank you for the response to the review. I'd like to clarify my point on Weakness 1, which the response did not adequately address. The point is not about specific hardware or system optimizations, but that the setup and the experiments in the paper do not generalize the proposed framework to evaluate agents that are based on different LLMs (e.g., one agent based on GPT-4o and another based on Claude Sonnet 4.5), or furthermore agent designs that leverage multiple different LLMs within a single agentic system. This is because token-based budgets are only fair and correlate with wall clock time when the same LLM is used everywhere in the experiments.
>
> That said, I remain positively leaning towards this paper's contributions.

---

### Official Review · Reviewer_TV6c · 2025-11-01

**Soundness:** 2
**Presentation:** 2
**Contribution:** 2
**Rating:** 4
**Confidence:** 3

**Summary:**

This paper addresses the limitation of existing LLM-based agents in dynamic environments (where environments evolve during agent reasoning) by formulating "real-time reasoning" as a new problem and building Real-Time Reasoning Gym (with Freeway, Snake, Overcooked) to evaluate it. It studies two agent paradigms—reactive agents (bounded computation for speed) and planning agents (extended reasoning for complexity)—finds their flaws, and proposes AgileThinker, a dual-thread model that combines both by letting the reactive thread use partial planning traces. Experiments show AgileThinker outperforms single-paradigm agents as cognitive load and time pressure increase, and wall-clock tests confirm the linear correlation between tokens and real inference time, verifying its practicality.

**Strengths:**

1. It innovatively defines the "real-time reasoning" problem for LLM agents (addressing the flaw that environments evolve parallel to agent reasoning) and builds Real-Time Reasoning Gym with three games (Freeway, Snake, Overcooked) to systematically control cognitive load and time pressure (using tokens as a hardware-agnostic time proxy).
2. The proposed AgileThinker has an innovative dual-thread design: its reactive thread references partial planning traces for real-time decisions, solving the limitations of existing dual-system methods (independent operation or waiting for completion), and experiments with the same model family verify its superiority over single-paradigm agents.
3. It emphasizes practicality: wall-clock experiments confirm a strong linear correlation between token count and real inference time (R²=0.9986).

**Weaknesses:**

1. AgileThinker lacks an adaptive mechanism for thread resource allocation: the optimal token budget for the reactive thread ( $N_{TR}$ ) varies across environments (e.g., ~5k tokens for Freeway vs. ~2k tokens for Snake/Overcooked) and requires manual empirical tuning, with no solution proposed to dynamically adjust it based on real-time environmental changes.
2. Experimental scenarios are disconnected from real-world complexity: all experiments are conducted on three simulated games (Freeway, Snake, Overcooked), and there is no verification of AgileThinker’s performance in real-world complex scenarios (e.g., SWE-Bench, BrowseComp), weakening the practical reference value of the results.
3. The contribution is unclear. Existing works [1] control whether models conduct deep thinking through mode switching; what is the essential difference between AgileThinker and these works?

[1] Wu, S., Xie, J., Zhang, Y., Chen, A., Zhang, K., Su, Y., & Xiao, Y. (2025). ARM: Adaptive Reasoning Model. ArXiv, abs/2505.20258.

**Questions:**

In terms of evaluation tools, what unique contributions does the Real-Time Reasoning Gym (with three games: Freeway, Snake, Overcooked) bring to the field of real-time agent evaluation?

---

> ### Author Response · Authors · 2025-11-20
>
> Thank you for your clear and helpful review! We appreciate your feedback and have addressed your points below.
>
> **W1**:
>
> In Section 5, we did extensive research on the internal budget hyperparameter and our insight is that *performance peaks when $N_{T_\mathcal{R}}$ approximates the natural upper bound of the reactive thread*. However, inspired by both your comments and the [AIMD algorithm](https://en.wikipedia.org/wiki/Additive_increase/multiplicative_decrease), we did additional experiments to show that **thread resource can be allocated adaptively** through a simple *dynamic adjustment mechanism* for $N_{T_\mathcal{R}}$.
>
> Specifically, we initialize $N_{T_\mathcal{R}}$ to a small value (1k tokens). If the reactive thread fails to generate a complete response within the allocated tokens, we increase $N_{T_\mathcal{R}}$ by 1k tokens. Conversely, if the reactive thread uses fewer than tokens for 3 consecutive responses, we decrease $N_{T_\mathcal{R}}$ by 0.5k tokens. This approach eliminates the need  to predefine a fixed $N_{T_\mathcal{R}}$, allowing the model to adapt it based on query complexity.
>
> This method turns out to be effective. Since the DeepSeek official API currently only supports DeepSeek-V3.2, we conducted additional experiments on this model (You can find more results in our revision), using "thinking" for planning and "non-thinking" for reactive with the adapter . The scores under **Hard** cognitive load and **8k tokens/step** time pressure are shown below:
>
> |  | Reactive | Planning | AgileThinker (fixed, optimal $N_{T_\mathcal{R}}$) | AgileThinker (adaptive $N_{T_\mathcal{R}}$) |
> |--------|---------|----------|---------------------|--------------------------|
> | Freeway | 0.4659|0.1025|0.6352 (3k)| 0.6279 |
> | Snake | 0.2556 | 0.0550 | 0.3456 (1k) | 0.4166 |
> | Overcooked | 0.5246 | 0.0134 | 0.6563 (2k)| 0.6741 |
>
>
> **W2**:
>
> We appreciate your concern regarding real-world applicability. We believe our conclusions **generalize to real-world deployment** since the settings we choose **cover a wide variety of cognitive loads, time pressure, and even different kinds of real-time dynamics** (hazards, opportunities, partners). This comprehensiveness gives a full-picture of the challenges of real-time reasoning and the advantage of AgileThinker.
>
> Moreover, we choose to use game environments for following reasons:
>
> (1) They provide a **controlled** setting where we can systematically vary difficulty and urgency, allowing us to isolate and study the effects of our architectural design. Games have long served as testbeds for novel AI problems,  with solutions later applied to application-driven tasks (e.g., interactive NLP was studied in text games, and RL  was studied on Atari games).
>
> (2) **Real-world complexity is multi-dimensional**: It involves both **contextual complexity** (code/web) and **dynamic complexity** (static/dynamic). We created Real-Time Gym because existing benchmarks (like SWE-Bench or  BrowseComp) often ignore dynamic complexity, such as users changing their minds or updating the code base  during an agent's execution loop. Our work is an initial step in studying this specific axis. We acknowledge that an ideal benchmark would consider all realistic dimensions, but this is beyond the scope of  this paper.
>
> **W3**:
>
> Our proposed AgileThinker differs from mode-switching and other adaptive reasoning approaches in a key  aspect: our work aims to **combine** planning and reactive reasoning within a single agent, while existing adaptive models try to **switch between** or **trade off** these modes.
>
> In Appendix C.3, we experiment with a commercial adaptive reasoning model, Gemini-2.5-Flash. When we set the thinking budget to -1 ("Planning only" column), the  model adapts to problem difficulty; when we set a positive thinking budget ("Planning only + budget control" column), it adapts to the budget. We show that both adaptive methods (adapt to budget or difficulty) still struggle on our benchmark, indicating that, at the current stage, such methods cannot fully exploit the benefits of both reasoning modes simultaneously.

---

> > ### Comment · Reviewer_TV6c · 2025-11-24
> >
> > Thanks for your response. I still have a question. In W3, what is the specific difference between **combine** and **switch**? Under the **combine** setting, it seems that it also need to switch between planning and reactive patterns?

---

> > > ### Author Response · Authors · 2025-11-24
> > >
> > > Thank you for your insightful question!
> > >
> > > The key distinction lies in how the models operate:
> > >
> > > - **Switch** setting: Only **one** model runs at any given time. The system selects which model type (planning or reactive) to use based on the current situation, switching between them as conditions change.
> > > - **Combine** setting (ours): **Both** models run in parallel to handle the situation simultaneously. Typically, the planning agent operates continuously throughout the entire time frame, while the reactive agent activates for shorter periods, as it requires less computational reasoning.
> > >
> > > We would like to draw your attention to Figure 4, which illustrates this concurrency in our design: in the combine setting, when the reactive agent is actively responding, the planning agent continues its reasoning process in the background. This parallel operation fundamentally distinguishes it from the switch setting, where only one agent is active at a time.
> > >
> > > We hope this clarifies the distinction. Please let us know if you have any further questions :)

---

> ### Author Response · Authors · 2025-11-22
>
> **Q1**:
>
> We are fully prepared to open-source the implementation of the evaluation framework, environment logic and agent architectures. It is an evaluation system that **supports all vllm compatible models**, with **flexible configurations** for cognitive load, time pressure, and time unit format (e.g., tokens/step, seconds/step). Moreover, it can be used not only for evaluation purposes but also for training new agents, as it provides **openai gym style APIs for environment-agent interaction**.
> It is also **extendable** since it's easy to add new environments and agent designs. We believe this gym will facilitate future research on real-time reasoning problems  and help in forstering the development of more real-time capable LLM agents.
>
> In summary, we believe we have done additional experiments to address the limitation you proposed in **W1** and clarified our method's practicability and unique contribution in the response to **W2** and **W3**. We would be very grateful if you could check these and reconsider the score if you think they have addressed your concerns.

---

### Author Response · Authors · 2025-12-03
**General Response to Area Chair**

We sincerely thank all reviewers for their thorough evaluations and constructive feedback. We are encouraged by the reviewers' recognition of our work's **novelty** (sdJh, EhbH), **strong motivation and presentation** (hZPr, sdJh, EhbH), and **comprehensive experimental validation** (all reviewers). We particularly appreciate that **two reviewers assigned a rating of 8 (accept)** and acknowledged the significance of introducing real-time reasoning as a critical new problem for LLM agents.

## Summary of Reviewer Reception

The reviewers consistently acknowledged:

- **Novel problem formulation**: Real-time reasoning in evolving environments addresses an under-explored but practically important challenge (hZPr, sdJh, EhbH)
- **Well-designed benchmark**: Real-Time Reasoning Gym provides a systematic testbed with controlled cognitive load and time pressure variables (all reviewers)
- **Strong empirical results**: Our simple yet effective AgileThinker design demonstrates clear advantages over single-paradigm agents, especially under high time pressure (sdJh, EhbH)
- **Clear presentation**: The paper effectively communicates complex concepts through well-structured writing and illustrations (sdJh, EhbH)

## Major Concerns Addressed

We have carefully addressed all major concerns raised during the review process:

### 1. Adaptive Thread Resource Allocation (TV6c-W1)

**Concern**: The optimal token budget $N_{T_R}$ requires manual tuning across different environments.

**Response**: In our original submission, we conducted extensive research on the internal budget hyperparameter and demonstrated that performance peaks when $N_{T_R}$ approximates the natural upper bound of the reactive thread. Building upon this finding and inspired by reviewer feedback, we developed a dynamic adjustment mechanism based on the AIMD algorithm that eliminates the need for environment-specific tuning.

Our experiments on DeepSeek-V3.2 show that adaptive $N_{T_R}$ achieves comparable or superior performance to manually tuned optimal values:

| Environment | Reactive | Planning | AgileThinker (fixed) | AgileThinker (adaptive) |
| ----------- | -------- | -------- | -------------------- | ----------------------- |
| Freeway     | 0.4659   | 0.1025   | 0.6352 (3k)          | **0.6279**              |
| Snake       | 0.2556   | 0.0550   | 0.3456 (1k)          | **0.4166**              |
| Overcooked  | 0.5246   | 0.0134   | 0.6563 (2k)          | **0.6741**              |

This adaptive strategy allows dynamic adjustment based on test-time observations and will be included in our revision.

### 2. Token-as-Time Proxy Validity (hZPr-W1)

**Concern**: Token budget only correlates with wall-clock time when using the same LLM, limiting generalizability across different models.

**Clarification**: We acknowledge this limitation and emphasize that:

- Our **Section 6 wall-clock experiments** naturally account for model-specific and hardware factors
- We deliberately use the same model family (DeepSeek) to isolate **architectural benefits** from confounding variables
- Our contribution focuses on **high-level agent system design** rather than low-level optimization
- The architectural principles we propose (parallel reasoning paradigms) generalize across different LLM implementations

### 3. Real-World Applicability (TV6c-W2, EhbH-W1)

**Concern**: Experiments limited to simulated game environments may not represent real-world complexity.

**Response**: We believe our findings generalize because:

1. **Comprehensive coverage**: Our settings span diverse cognitive loads, time pressures, and real-time dynamics (hazards, opportunities, cooperation)
2. **Controlled experimentation**: Games provide systematic control of difficulty and urgency, allowing us to isolate architectural effects—a paradigm successfully used in prior AI research (text games for interactive NLP, Atari for RL)
3. **Multi-dimensional complexity**: Real-world scenarios involve contextual complexity (code/web), dynamic complexity (static/dynamic), and other dimensions. We specifically address the under-explored **dynamic complexity axis**. We acknowledge that an ideal benchmark would consider all realistic dimensions, but this is beyond the scope of this paper.
4. **Extensibility**: Our architecture naturally extends to multi-modal and continuous-control tasks as LLM inference capabilities improve

---

> ### Author Response · Authors · 2025-12-03
>
> ### 4. Comparison with Existing Work (hZPr-W3, EhbH-W2, TV6c-W3)
>
> **Concern**: Need better contextualization with related work, particularly adaptive reasoning models and dual-system agent designs like Parallelized Planning-Acting (Li et al., 2025).
>
> **Response**: We have comprehensively addressed this concern:
>
> **(a) Dual-system agent baselines**: We conducted additional experiments with Li et al.'s Parallelized Planning-Acting method. Our analysis shows it falls into the category of **planning-only baseline with minor variations**, as its plan thread cannot take effect until completion. In contrast, AgileThinker's reactive thread immediately leverages partial planning traces:
>
> | Method           | Freeway    | Snake      | Overcooked |
> | ---------------- | ---------- | ---------- | ---------- |
> | Reactive         | 0.4659     | 0.2556     | 0.5246     |
> | Planning         | 0.1025     | 0.0550     | 0.0134     |
> | **AgileThinker** | **0.6352** | **0.3456** | **0.6563** |
> | Li et al. [1]    | 0.2022     | 0.0083     | 0.0115     |
>
> **(b) Adaptive reasoning models**: We clarified the fundamental difference between AgileThinker and mode-switching approaches. In Appendix C.3, we demonstrate that commercial adaptive models (e.g., Gemini-2.5-Flash) still struggle in our benchmark, as they switch between or trade off reasoning modes rather than **combining both paradigms simultaneously**.
>
> **(c) Enhanced related work**: We have expanded our discussion to include:
>
> - **Non-LLM real-time agents**: Added review of classical real-time AI research (best-first search, robotics) to better contextualize our contributions
> - **Event-driven vs. computational-latency perspectives**: Clarified that prior work focuses on asynchronous event management, while we address the LLM-specific challenge of balancing extended reasoning computation against timely responses
>
> ### 5. Additional Clarifications (sdJh Questions)
>
> We addressed all presentation questions from Reviewer sdJh, including clarifications on:
>
> - Figure labeling (Reference inputs, agent outputs)
> - Score normalization methodology (detailed in Appendix A)
> - Code-as-policy categorization as planning (conditional logic enables adaptability)
> - Case study authenticity (all trajectories are real)
> - Experimental settings (Medium cognitive load for Table 2)
>
> ## Revisions and Improvements
>
> Based on reviewer feedback, our revision includes:
>
> 1. **New experiments**: Adaptive $N_{T_R}$ mechanism validation (TV6c)
> 2. **Baseline comparisons**: Comprehensive evaluation against dual-system designs (EhbH, TV6c)
> 3. **Expanded related work**: Better positioning within both classical and modern real-time agent literature (hZPr)
> 4. **Clarified presentation**: Addressed all figure labeling and methodology questions (sdJh)
> 5. **Code release commitment**: Open-sourcing Real-Time Reasoning Gym with vllm-compatible implementation, OpenAI Gym APIs, and extensible environment/agent interfaces (sdJh)
>
> ## Core Contributions Reaffirmed
>
> Our work makes three fundamental contributions to the field:
>
> 1. **Problem formulation**: First to systematically study LLM agent reasoning under continuous environmental evolution with hardware-agnostic time pressure control
> 2. **Benchmark design**: Real-Time Reasoning Gym provides a reproducible, extensible testbed covering diverse real-time dynamics and difficulty levels
> 3. **Architectural innovation**: AgileThinker's simple yet effective parallel dual-thread design achieves superior balance between reasoning depth and response latency, with performance validated across multiple models and settings

---

### Meta-Review · Area_Chair_vEAt · 2026-01-08

**Summary:**

The paper proposes "real-time reasoning" as a distinct problem setting where the environment continues to evolve while the agent is performing inference. To study this, the authors introduce the Real-Time Reasoning Gym (using environments like Freeway, Snake, and Overcooked) and propose "AgileThinker," a dual-thread architecture combining a reactive module with a parallel planning module.

**Reviewer Concerns:**

**Addressed:**

**Baselines and Related Work:** Reviewer EhbH requested a comparison to "Parallelized Planning-Acting" (Li et al., 2025). The authors performed this comparison, showing AgileThinker outperforms it because the baseline waits for plans to finalize before acting, whereas AgileThinker's reactive thread acts immediately.


**Hyperparameter Tuning:** Reviewer TV6c raised concerns about the manual tuning required for the token budget (). The authors addressed this by implementing an adaptive mechanism based on the AIMD algorithm, which achieved comparable performance to the manually tuned version.


**Clarification on "Real-Time":** Reviewer hZPr noted that the actual wall-clock time is in minutes due to API latency, not milliseconds. The authors acknowledged this infrastructure limitation but argued the architectural principles for handling time pressure hold regardless of the specific timescale.



**Outstanding:**

**Token-as-Time Proxy:** Reviewer hZPr raised a valid point that using token counts as a proxy for time only works when comparing the exact same model, as different models (e.g., GPT-4 vs Claude) have drastically different latencies per token. While the authors argue they control for this by using the same model family, this remains a limitation for the benchmark's generalizability across different providers.


**Simulated vs. Real World:** Reviewers TV6c and EhbH noted that simulated games may not fully capture the complexity of real-world dynamic tasks (like robotics).

**Reviewer Scores:**

**Reviewer TV6c (Current: 4):** will be 6. The authors addressed their main technical concern regarding the adaptive token budget and clarified the difference between mode-switching and the proposed parallel architecture.


**Reviewer hZPr (Current: 6):** wont change but should be more positive (still 6). This reviewer explicitly stated in the discussion that they "remain positively leaning towards this paper's contributions" despite the outstanding concern about generalizability across different LLMs.

**Reviewer sdjh (Current: 8):** wont change 8. They remained positive throughout and kept their score.

**Reviewer EhbH (Current: 8):** wont change 8. They were satisfied with the baseline comparisons and maintained their score.

---

### Decision · Program_Chairs · 2026-01-26

Accept (Poster)